# Learning Biophysical Models of Large-Scale Multineuronal Data to Enable Precise Neurostimulation

**Amrith Lotlikar** [1][*]  **Ian Christopher Tanoh** [2][*]  **Praful Vasireddy** [1]  **Andrew Lanpouthakoun** [2]
**Ramandeep Vilkhu** [1]  **Michael Sommeling** [1]  **A.J. Phillips** [1]  **Alexander Sher** [3]  **Alan Litke** [3]
**Scott W. Linderman** [2][†]  **E.J. Chichilnisky** [4][†]  **Subhasish Mitra** [1][†]

## Abstract

Multi-compartment Hodgkin–Huxley (HH) models provide a principled framework for predicting neural dynamics and responses to electrical stimulation. However, fitting HH biophysical parameters typically requires intracellular recordings, which are invasive and low-throughput, limiting the ability to capture the geometry and cell-specific properties of many neurons in a given neural circuit. Multi-electrode arrays (MEAs) offer a scalable alternative—high-density extracellular measurements from full neural populations—but HH model complexity has so far precluded reliable biophysical inference from extracellular data alone. Here, we introduce a framework to rapidly infer HH parameters from designed features of extracellular MEA measurements by leveraging differentiable biophysical simulation and simulation-based inference, unlocking a wide range of downstream applications. In this work, we focus on a central goal of translational neuroengineering: predicting neural spiking responses to candidate neurostimulation patterns that would take hours to measure clinically. To validate our approach, we collected hundreds of hours of stimulation and recording data from isolated macaque retina with a 30 μm-pitch 512-electrode array. Our framework predicted previously unseen multi-electrode stimulation responses with 90.4% accuracy using HH models fit from only a few minutes of recording, replacing hours of stimulus testing.

*Indicates lead authors (contributions in Appendix A), †Equal supervisor contribution. [1]Department of Electrical Engineering, Stanford University, Stanford, CA, USA [2]Department of Statistics, Stanford University, Stanford, CA, USA [3]Santa Cruz Institute for Particle Physics, University of California, Santa Cruz [4]Department of Neurosurgery, Stanford University, Stanford, CA, USA. Correspondence to: Amrith Lotlikar <lotlikar@stanford.edu>.

*Proceedings of the 43rd International Conference on Machine Learning*, Seoul, South Korea. PMLR 306, 2026. Copyright 2026 by the author(s).

## 1. Introduction

Hodgkin–Huxley (HH) models (Hodgkin & Huxley, 1952) form the basis of our understanding of neural computation at the cellular level. By providing a biophysically grounded inductive bias for cell-specific fitting with limited data, HH models could unlock many applications. For example, they could be used to capture the function of large neural circuits and design precise neurostimulation patterns to elicit desired responses.

However, current approaches for fitting HH models face several challenges. Typically, single-neuron HH models are fit to intracellular recordings that provide direct measurements of transmembrane voltage. However, such measurements can only be performed with one or a few electrodes at a time, precluding their use for modeling every neuron in a large population. Extracellular multi-electrode recordings can provide spiking information about many neurons simultaneously, but these signals reflect the superimposed activity of multiple neurons near each electrode. Fitting HH models to such data is challenging due to parameter degeneracy: many different combinations of HH model conductances, morphologies, and electrode positions can produce similar recorded voltage dynamics. These challenges have prevented the development and use of accurate HH models of large neural populations for applications such as neuroengineering of implanted devices.

Here, we present a fast and biophysically grounded approach for inferring cell-specific multi-compartment Hodgkin–Huxley models from multi-electrode array (MEA) recordings. We leverage large-scale, high-density extracellular MEA recordings from isolated macaque retinas, as shown in Figure 1. To separate single-neuron signals from population extracellular data, we spike sort recordings of light-evoked retinal activity (Pachitariu et al., 2016) to extract each neuron's electrical image (EI) – the average multielectrode voltage waveform recorded during a spike. To alleviate the parameter degeneracy problem, we design biophysically interpretable EI features (e.g., spike duration and propagation velocity) and develop a method to incorporate a small number of easily measured stimulation features

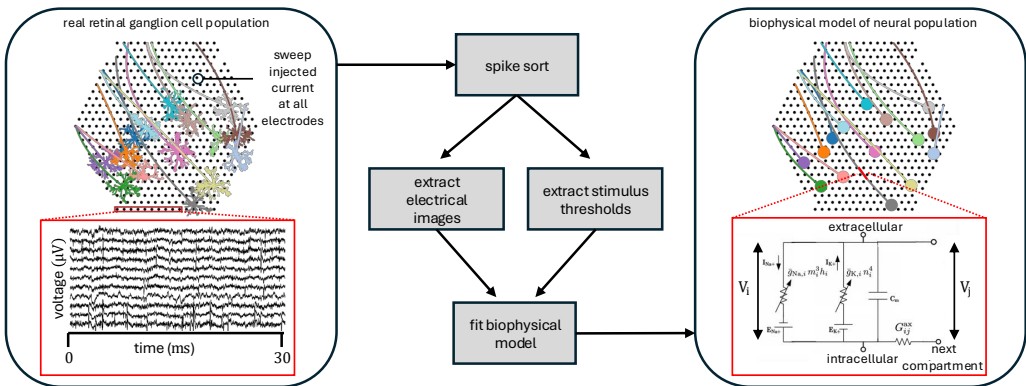

*Figure 1.* Overview of our digital twin framework. **Left:** A population of RGCs recorded on a high-density multi-electrode array shown schematically. Inset: extracellular voltage traces recorded during light-evoked neural activity. **Center:** spike sorting extracts electrical images and stimulus thresholds, which are used to fit biophysical models. **Right:** The resulting biophysical model of the neural population, with somas (colored circles) and axons (colored lines) positioned relative to the electrode array (black dots). The RGC axon is modeled as compartments connected to form a cable. Inset: equivalent circuit for a single compartment, showing membrane capacitance, voltage-gated ion channels, intracellular potential $V_i$, and axial conductance $G^{\text{ax}}$ coupling to adjacent compartments.

(single-electrode thresholds) to constrain inference. Finally, to tractably fit complex biophysical models at scale, we employ gradient-based optimization in the differentiable simulator JAXLEY (Deistler et al., 2025b) and simulation-based inference (Cranmer et al., 2020).

Models fit to only minutes of extracellular MEA retinal recordings predicted unseen responses to simultaneous multi-electrode stimulation with 90.4% accuracy, replacing hours of stimulus testing. The results indicate that this approach provides an efficient new way to fit biophysical models to many neurons in a neural circuit simultaneously, and that the resulting simulation can provide valuable predictions for engineering a neural implant. More broadly, the ability to produce a biophysically accurate *digital twin* of a neural circuit with readily available data may have many scientific and engineering applications. Data and code for our approach is available at `https://github.com/amrith1/rgc-ei-fit-hh`.

## 2. Background

This section reviews Hodgkin-Huxley models and describes how they relate to the measured extracellular voltage recordings and predict responses to electrical stimulation.

### 2.1. Action Potentials (Spikes) and the Electrical Image

Action potentials, or *spikes*, are the primary events detected in extracellular recordings. When a neuron spikes, it adds a stereotyped spatiotemporal waveform to the multielectrode voltage recording – we refer to this waveform as the *electrical image* (EI; see fig. 2). Although extracellular recordings of dense neural circuits such as the retina produce superimposed spikes from many cells, spike-sorting

methods such as Kilosort2 (Pachitariu et al., 2016) can infer both the number of neurons present and each neuron's EI by unsupervised clustering of repeated spike waveforms in recorded MEA voltages (Figure 1), leveraging the consistency of each neuron's EI across spikes. Thus, the signature of each neuron's spiking is revealed by its EI. All EIs were obtained from recordings of light-evoked retinal activity. At each electrode, the EI waveform typically consists of three peaks corresponding to capacitive, sodium, and potassium phases (Figure 2B); we name these peak features following the convention of Gold et al. (2006).

### 2.2. Hodgkin–Huxley model

We model the electrical activity of neurons using the multi-compartment HH formulation. Each neuron is represented as a collection of electrically connected cylindrical membrane compartments indexed by $i$, each with time-varying membrane voltage $V_i(t)$ and ion channel gating variables $m_i(t)$, $h_i(t)$, and $n_i(t)$ that govern transmembrane ionic currents. The equivalent circuit summarizing the voltage dynamics for a single compartment is shown in Figure 1 (right).

**Fitting Objective** From electrical images, we estimate a neuron-specific parameter vector, $\boldsymbol{\theta} = \{\bar{g}_{\text{Na},i}, \bar{g}_{\text{K},i}, r_i, \mathbf{x}_i\}_i$, where $\bar{g}_{\text{Na},i}$ and $\bar{g}_{\text{K},i}$ are maximal sodium and potassium conductance densities, $r_i$ are segment radii, and $\mathbf{x}_i \in \mathbb{R}^3$ are segment locations. All remaining biophysical parameters, ion channel kinetics, and initial conditions for $V_i, m_i, h_i$ and $n_i$ are assumed to be known constants that are common across retinas based on prior anatomical and intracellular neural recording studies (Fohlmeister et al., 2010; Kish et al., 2023; Vilkhu et al., 2025). Constant values for each fitting

experiment are reported in Appendix L.

In HH models, the membrane voltage of each compartment evolves according to

$$S_i C_m \frac{dV_i}{dt} = I_i^{\text{stim}}(t) - S_i J_i^{\text{ion}} + \sum_{i\,j} G_{ij}^{\text{ax}} (V_j - V_i) \tag{1}$$

where $S_i = 2\pi r_i l_i$ is the compartment surface area, $l_i$ is the compartment length, $C_m$ is the membrane capacitance per unit area, $I_i^{\text{stim}}(t)$ is an applied stimulus current, $J_i^{\text{ion}}$ is the transmembrane ionic current density, and $G_{ij}^{\text{ax}}$ is the axial conductance coupling segment $i$ to segment $j$. $G_{ij}^{\text{ax}}$ is a function of compartment radii and lengths (see Appendix B).

The ionic current density is given by

$$J_i^{\text{ion}} = \bar{g}_{\text{Na},i}\, m_i^3 h_i\, (V_i - E_{\text{Na}}) + \bar{g}_{\text{K},i}\, n_i^4\, (V_i - E_{\text{K}}) \tag{2}$$

where $E_{\text{Na}}$ and $E_{\text{K}}$ are ionic Nernst potentials. The gating variables $m_i$, $h_i$, and $n_i \in [0,1]$ represent the fraction of open ion channels and evolve according to standard first-order kinetics

$$\frac{dz_i}{dt} = \alpha_z(V_i)\,(1 - z_i) - \beta_z(V_i)\, z_i, \quad z_i \in \{m_i, h_i, n_i\}, \tag{3}$$

with measured voltage-dependent rate functions $\alpha$ and $\beta$ for each gating variable. Together, equations (1)–(3) define a nonlinear dynamical system in which action potentials are produced and propagate into neighboring compartments through axial coupling.

### 2.3. Forward Model 1: Electrical Image

To define a model for computing neural electrical images to be used in biophysical parameter inference, we introduce

$$f_{\text{EI}} : (\boldsymbol{\theta}, \{\mathbf{x}_m\}_m) \mapsto \{\Phi_m(t)\}_m,$$

which maps the neuron-specific parameter vector $\boldsymbol{\theta}$ and electrode locations $\{\mathbf{x}_m\}_{m=1}^{M} \subset \mathbb{R}^3$ to the extracellular voltages recorded across electrodes during a single action potential. Given $\boldsymbol{\theta}$ and the Hodgkin–Huxley dynamics defined above, we simulate an action potential by initializing $V_i(0)$ at designated initiation segments (Appendix C) to a critical voltage that initiates a spike. The resulting spike propagates along the neuron according to the HH dynamics, which we integrate forward in time using JAXLEY, a differentiable simulator for multi-compartment HH models (Deistler et al., 2025b). Exact initial conditions used in experiments are reported in Appendices I and L. The extracellular potential at electrode $m$ is then computed from the HH dynamics using the line-source approximation (Gold et al., 2006):

$$\Phi_m(t) \approx \frac{1}{4\pi\sigma} \sum_i \frac{S_i \big( J_i^{\text{ion}}(t) + C_m \frac{dV_i}{dt} \big)}{\| \mathbf{x}_i - \mathbf{x}_m \|}, \tag{4}$$

where $\sigma$ is the extracellular conductivity, $\mathbf{x}_i$ is the location of membrane compartment $i$, and $\mathbf{x}_m$ is the electrode location. The collection of waveforms recorded on different electrodes $\{\Phi_m(t)\}_m$ produced by a single simulated action potential constitutes the electrical image associated with $\boldsymbol{\theta}$. An example of a simulated axonal electrical image is shown in Figure 2B, reproducing the characteristic three-phase electrical image observed near the axons of real neurons.

### 2.4. Forward Model 2: Stimulation

The electrodes of an MEA can be used to inject current into the extracellular medium to elicit neural spikes. To predict whether such a stimulus induces a spike, we define a second forward model

$$f_{\text{stim}} : (\boldsymbol{\theta}, \{\mathbf{x}_m, I_m(t)\}_m) \mapsto \{0, 1\},$$

which maps biophysical parameters $\boldsymbol{\theta}$, electrode locations $\{\mathbf{x}_m\}_m$, and electrode-indexed injected current waveforms $\{I_m(t)\}_m$ to a binary spike/no-spike outcome. Given these inputs, the model integrates the Hodgkin–Huxley dynamics forward in time and outputs whether an action potential is induced. The full construction of the stimulation forward model, including how injected currents enter the membrane dynamics, is provided in Appendix C.

## 3. Learning Biophysical Models

We seek to estimate parameters $\boldsymbol{\theta}$ for each neuron so the outputs of models $f_{\text{EI}}$ and $f_{\text{stim}}$ match observed electrical images and stimulation outcomes. Then, we can use estimated parameters, $\hat{\boldsymbol{\theta}}$, and the forward model, $f_{\text{stim}}$, to predict previously unseen stimulus responses. We consider two separate approaches to estimate $\boldsymbol{\theta}$: gradient descent and simulation-based inference, which share a common feature extraction step.

### 3.1. Feature Extraction

**Electrical Image Features** Because electrical images (EIs) are high-dimensional spatiotemporal signals that vary in amplitude, duration, and relative timing across electrodes, directly comparing simulated and experimentally recorded EIs with pointwise losses is unreliable and sensitive to temporal misalignment. To enable accurate and stable estimation of biophysical parameters, we designed a set of physically interpretable EI features that capture key aspects of spike shape and propagation while varying smoothly with respect to $\boldsymbol{\theta}$. These features are inspired by prior analyses of extracellular spike waveforms (Gold et al., 2007) and are implemented in a fully differentiable manner to support gradient-based optimization. At each electrode, the EI waveform is highly stereotyped and exhibits three prominent peaks corresponding to capacitive, sodium, and potassium phases, following established conventions (Gold

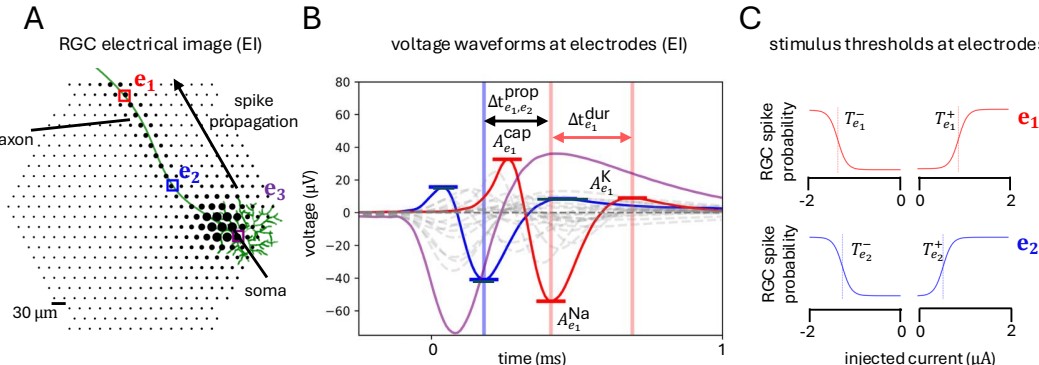

*Figure 2.* Features of extracellular recording and stimulation. (A) Kilosort2-derived electrical image (EI) of an individual retinal ganglion cell recorded on a 512-channel, $30\mu m$-pitch MEA. Black circles denote electrodes, with area proportional to EI signal strength. (B) EI waveforms at many representative electrodes. Five differentiable EI features are used for fitting: three per-electrode peak amplitudes—capacitive ($A^{\mathrm{cap}}$), sodium ($A^{\mathrm{Na}}$), and potassium ($A^{\mathrm{K}}$)—corresponding to stereotyped phases of the spike that reflect underlying $Na^+$ and $K^+$ transmembrane currents; and two timing features—spike duration ($\Delta t^{\mathrm{dur}}$) between the sodium and potassium peaks (per electrode) and propagation delay ($\Delta t^{\mathrm{prop}}$) of the sodium peak across electrode pairs (a proxy for axonal conduction velocity). (C) Stimulus thresholds ($T_m^+, T_m^-$) for each electrode, defined as the injected current amplitude at which the cell spikes with $> 50\%$ probability in experiment, for anodic and cathodic pulses, respectively.

et al., 2006). Specifically, we extract per-electrode peak amplitudes for the capacitive, sodium, and potassium components ($A^{\mathrm{cap}}, A^{\mathrm{Na}}, A^{\mathrm{K}}$). We extract action potential duration ($\Delta t^{\mathrm{dur}}$), defined as the time between the sodium and potassium peak, and pairwise electrode sodium-peak timing differences ($\Delta t^{\mathrm{prop}}$), which together serve as a proxy for spike propagation velocity. A qualitative illustration of these features is shown in Figure 2, with mathematical definitions provided in Appendix E.

**Stimulus thresholds.** To further constrain biophysical parameters, we incorporate neural stimulus thresholds as additional features derived from extracellular stimulation experiments. For a fixed electrode $m$, the probability that an applied current induces an action potential varies monotonically with stimulus intensity, implying a well-defined positive and negative threshold, ($T_m^+, T_m^-$), at which the neuron spikes with $50\%$ probability (Figure 2C). The thresholds measured at different electrodes reveal the sensitivity of the neuron to electrical stimulation at different locations with respect to the cell. Given a parameter vector $\boldsymbol{\theta}$, stimulus thresholds can be computed using the forward stimulation model $f_{\mathrm{stim}}$ by evaluating whether an action potential is induced across a range of stimulus amplitudes and identifying the lowest amplitude at which spiking occurs. However, since $f_{\mathrm{stim}}$ produces a binary output, the resulting threshold is not differentiable with respect to $\boldsymbol{\theta}$. To enable gradient-based optimization, we introduce a novel differentiable relaxation $P_{\mathrm{stim},m}(\boldsymbol{\theta}, I)$ that maps single-electrode stimulation at an amplitude $I$ to a smooth probability of spike initiation in [0, 1] via a sigmoidal transformation. The detailed construction of $P_{\mathrm{stim},m}$ is described in Ap-

pendix D. During fitting, parameters are optimized so that $P_{\mathrm{stim},m}(\boldsymbol{\theta}, T_m^+) = 0.5$ (and analogously for $T_m^-$), enforcing consistency between predicted and measured stimulus thresholds.

### 3.2. Parameter Inference Methods

**Parameter Inference with Gradient Descent** Given observed electrical images and stimulus thresholds, our first method estimates biophysical parameters $\boldsymbol{\theta}$ by minimizing a differentiable loss $\mathcal{L}(\boldsymbol{\theta})$ that compares measured EI features and stimulus thresholds to forward-model predictions (defined Appendix F). Because we engineered EI timing features to be differentiable using soft argmin/max operators, gradients of $\mathcal{L}$ can be computed efficiently via automatic differentiation with JAXLEY. We optimize $\boldsymbol{\theta}$ using Adam gradient descent (Kingma & Ba, 2014), reparameterizing conductances, radii, and spatial locations with bounded transforms to enforce physiological constraints. Initial guesses of biophysical parameters are taken to be the midpoint of these physiological constraints. For RGCs, we initialize our guess of compartment locations with an EI driven method described in Appendix P. This combined procedure yields a point estimate of $\boldsymbol{\theta}$ that matches the observed recordings under the forward models. The full loss definition and implementation are provided in Appendix F.

**Simulation-Based Inference** While gradient-based fitting yields efficient point estimates of biophysical parameters, it assumes a deterministic forward model and therefore does not explicitly account for noise or model misspecification. To address this limitation, we also explore a stochastic for-

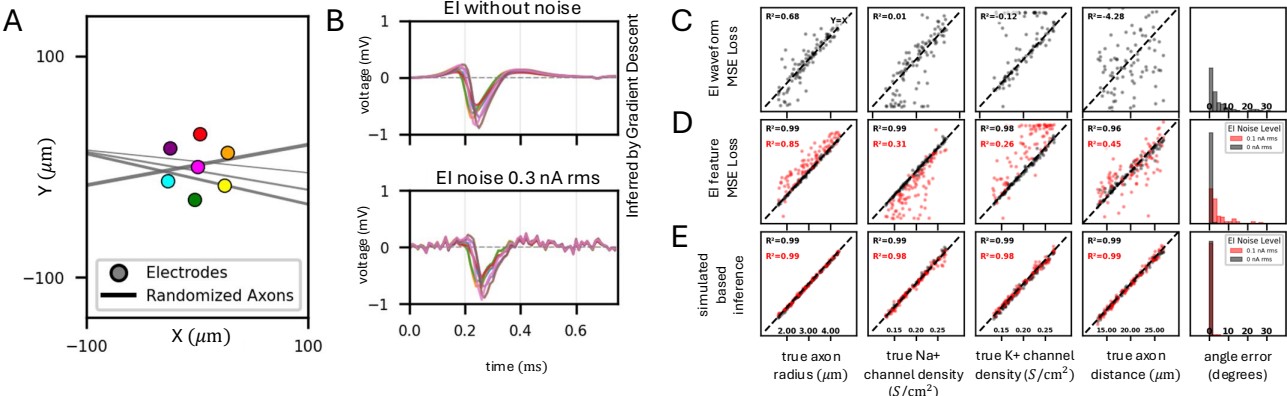

*Figure 3.* **Inference performance on noisy straight-axon simulations.** (A) Schematic of randomized straight axons (gray lines) relative to a 7-electrode array patch (colored circles). (B) Example EIs simulated without noise (top) and with forward-simulation noise (0.3 nA rms per-compartment current; bottom). (C) Inferred parameters plotted against ground truth for different inference methods: top row, gradient descent with waveform MSE loss (Appendix G); middle row, gradient descent with our differentiable feature-based loss; bottom row, simulation-based inference (SBI). The feature-based loss and SBI yield substantially improved parameter recovery. Rightmost column: axon trajectory angular error histograms. $R^2$ values denote coefficient of determination per method and parameter.

ward model by augmenting the Hodgkin-Huxley dynamics with additive Gaussian noise in the injected stimulus current $I^{\mathrm{stim}}$ in Equation 1, capturing variability in stimulation and unmodeled biophysical effects (Tanoh et al., 2026; Vilkhu et al., 2025). In principle, inference under this stochastic simulator could still be performed by optimizing the expected loss with respect to the injected-current noise using gradient descent; however, this would require high-dimensional integration over noise realizations and would yield only point estimates without calibrated uncertainty over parameters. We therefore perform inference under this stochastic simulator using simulation-based inference (SBI). SBI provides a Bayesian alternative for models with tractable simulators but intractable likelihoods, by learning an approximate posterior over parameters from simulated parameter–observation pairs. Specifically, we employ neural posterior estimation (Deistler et al., 2025a) to learn an approximate posterior over biophysical parameters $\boldsymbol{\theta}$ and an auxiliary noise parameter $\sigma_I$, representing the standard deviation of the injected current noise in Equation 1, given observed EI features and stimulus thresholds. Training data are generated by sampling $(\boldsymbol{\theta}, \sigma_I)$ from uniform priors over physiologically plausible ranges, simulating the stochastic forward models with additive Gaussian current noise of variance $\sigma_I^2$, and extracting the same features used for gradient-based fitting. A neural density estimator is trained on these simulated parameter–observation pairs to amortize inference. Once trained, the model enables rapid posterior sampling for new observations without further simulation, yielding calibrated uncertainty estimates over ion channel densities and neural geometry.

## 4. Experiments

**Overview** To evaluate HH parameter estimation from extracellular measurements, we performed simulated and real-data experiments, organized to progressively relax modeling assumptions. We first establish identifiability in a controlled setting by fitting straight, homogeneous axons and show that extracellular features and stimulation thresholds jointly identify the geometry and membrane conductances of the axon—a larger parameter set than in prior simulated extracellular inference work (Tanoh et al., 2026; Gold et al., 2007). We then validate the same inference machinery on simulated RGCs with realistic morphologies and physiologically constrained parameters. Finally, we applied the method to real RGC recordings and showed that fitted HH models generalize beyond the training condition by accurately predicting responses to previously unseen multi-electrode stimulation patterns.

### 4.1. Simulated Experiments

**Action potential simulations with a straight axon.** Biophysical parameter inference was first validated in a controlled setting by simulating action potentials propagating along straight axons positioned near a small 30 $\mu$m-pitch, 7-electrode patch (Figure 3A), representative of one electrode and its neighbors in the 512-electrode array used for recording (Figure 2). In a parameter sweep, the axon 3D position and orientation relative to the patch were randomized, together with axon radius and homogeneous Na$^+$/K$^+$ conductance densities, while all other biophysical and tissue parameters were held fixed. Axon geometry was parameterized by two control points in $\mathbb{R}^3$ (Appendix I), with all compartment locations constrained to lie on the line segment

between them. For each sampled parameter set, a synthetic electrical image (EI) was generated and stimulus thresholds were computed at each electrode (Figure 3B; Eq. (4)). This procedure yielded a dataset of paired ground-truth parameters and extracellular features. The setup is fully described in Appendix I.

**Feature-based losses make biophysical inference tractable.** To evaluate whether biophysical parameters could be recovered from extracellular measurements with gradient-based fitting, the HH model was fitted to each simulated EI under two objectives: (i) a waveform-level MSE—arguably the most direct and intuitive similarity metric, which compares EI waveforms after temporal alignment to handle variable EI duration from spike sorting (Appendix G)—and (ii) a differentiable MSE over EI features and stimulation thresholds (Appendix F; Section 3.2). Despite its intuitive appeal, minimizing waveform MSE yielded unreliable recovery and failed to identify key geometric parameters, most notably the axon-to-array distance (Figure 3C; $R^2 = -4.28$), motivating the need for a feature-based objective. By contrast, optimizing the feature-based loss enabled accurate recovery of geometry, radius, and conductance parameters across the same Monte Carlo distribution (Figure 3D). Finally, as a key contribution, we perform a systematic feature ablation study to identify which terms are necessary for successful inference (Appendix J); this analysis indicates that peak-amplitude features and pairwise electrode propagation-delay features are most critical for robust parameter recovery.

**Robustness to model misspecification with SBI.** To assess robustness to a controlled form of model misspecification, we introduced stochastic noise into the injected membrane current and evaluated parameter recovery across increasing noise levels using the stochastic forward simulator described in 3.2. Under this perturbation, gradient-based optimization degraded rapidly and exhibited systematic bias as expected since it optimized parameters with respect to the deterministic model (e.g., axon radius was overestimated while sodium channel density was underestimated; Figure 3). By contrast, SBI enables robust inference under a known stochastic forward model. When the posterior mean is taken as the SBI point estimate, recovery accuracy substantially exceeds that of Adam-based point estimates across noise regimes. Moreover, posterior uncertainty increases with noise level but remains small relative to the scale of the inferred parameters, indicating stable and well-calibrated inference (Appendix K). This suggests that a well-specified stochastic forward model could provide better estimates of biophysical parameters in real recordings, though here we only investigated noisy membrane currents.

**HH model for mammalian retinal ganglion cells.** To model RGCs with a more realistic geometry, a heterogeneous multi-compartment morphology with region-wise parameters was adopted from prior work (Vilkhu et al., 2025; Kish et al., 2023). Each RGC was partitioned into five connected regions—soma, axon hillock, sodium channel band (SOCB), narrow region (NR), and distal axon (DA) (Figure 11). Within each non-DA region, a fixed length was assumed and $Na^+/K^+$ conductance densities and radius were taken to be homogeneous; the NR radius was linearly tapered from the SOCB radius to the DA radius. Additional channel types (e.g., inward-rectifying K and Ca currents) were included but held fixed, since they weakly affect EI and stimulation responses in our regime (Vilkhu et al., 2025). Full details are in Appendix L.

**RGC compartment location parameterization.** To enforce smooth, but not straight, physically plausible axonal geometry, axon compartment locations were parameterized by a low-dimensional set of 3D control points. Compartment locations were obtained by B-spline interpolation through the control points (De Boor, 1986), with the first control point fixed at the soma. Interpolations were computed in jax using splinex (Braun, 2024). An example spline mapping of control points to compartment locations is in Appendix P.

**Simulated RGC recovery experiments.** To test parameter recovery in a heterogeneous setting, Monte Carlo experiments were performed with simulated RGCs analogous to the straight-axon sweep (Figure 3). Region-specific conductances and radii were sampled from physiologically motivated bounds, and control points were sampled from a fixed prior to generate diverse smooth trajectories (Appendix L). For SBI, the full parameter prior was provided; for gradient descent, control points were initialized at the prior mean. Across simulated cells, both gradient descent and SBI recovered conductances, radii, and control-point geometry accurately, with some degradation as compared to the straight axon experiment; full results in Appendix M.

### 4.2. Experiments with Recordings from Retinal Ganglion Cells

To evaluate the performance of the approach in a real neural circuit, HH models were fitted to RGCs recorded in isolated macaque retina with a multi-electrode array. A summary of tissue sourcing details and wet lab experimental protocols is provided in Appendix N. The same inference pipeline as in the simulated RGC experiments was used, except that compartment locations were initialized from a data-driven estimate of soma position and axon trajectory computed from the recorded EI (Appendix P); this estimate was used to initialize gradient-based optimization and to center the

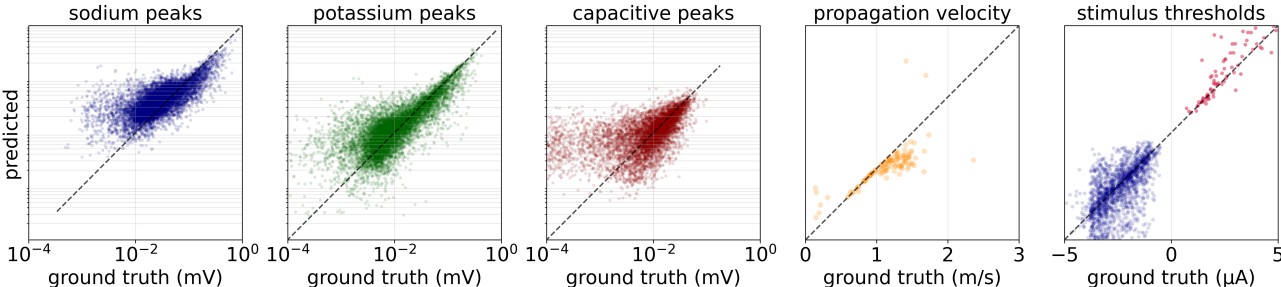

*Figure 4.* **Model predictions of training features on real RGC recordings.** For each of 198 parasol cells, Hodgkin–Huxley model parameters $\boldsymbol{\theta}$ were inferred from EI features and single-electrode thresholds. A predicted EI was generated with $f_{\mathrm{EI}}(\boldsymbol{\theta})$ and re-processed using the same feature extractor. Scatter plots compare predicted versus observed features across five extracted quantities: sodium, potassium, and capacitive peak amplitudes (log–log axes), propagation velocity, and stimulus threshold (linear axes). Feature comparisons were restricted to electrodes within $45\,\mu m$ of the fitted morphology. $R^2$ scores for each feature: sodium 0.614, potassium 0.760, capacitive 0.372, velocity 0.406, and thresholds 0.522. Additional analyses for the duration feature and for SBI-inferred models, which suffered from greater misspecification, are provided in Appendix Q.

SBI prior. Because biophysical ground truth is unavailable in real tissue, evaluation was performed indirectly on 198 parasol cells recorded in 13 retinal preparations, including 37 cells with measured multi-electrode stimulation responses. Properties of the full dataset are summarized in Appendix O. In contrast to the simulated experiments where the forward model was known exactly, the forward model is misspecified for real electrical images and stimulus responses. Model misspecification was characterized by how well fitted models reproduced the engineered EI features and single-electrode threshold constraints used for fitting. Generalization was then evaluated by predicting spike/no-spike outcomes for previously unseen multi-electrode stimulation patterns.

**Models fitted to real RGC measurements.** To assess model specification on real macaque recordings, where biophysical ground truth is unavailable, fitted HH models were evaluated by how well their forward predictions reproduced the extracellular constraints used for learning. For each cell, parameters $\boldsymbol{\theta}$ were inferred from EI features and single-electrode thresholds; a model-predicted EI was generated with $f_{\mathrm{EI}}(\boldsymbol{\theta})$ and re-processed with the same feature extractor to compare predicted versus observed features. Because EIs are spatially sparse, this analysis was restricted to electrodes within $45\,\mu m$ of the fitted morphology. Figure 4 summarizes predicted-versus-observed scatter plots for the primary supervision signals used in fitting (sodium and potassium peak amplitudes, propagation-delay features, and stimulus thresholds). The duration feature (which was substantially more misspecified in real tissue) and analogous plots for SBI-fitted models (which followed similar trends but were slightly degraded relative to gradient-based fits) are deferred to Appendix Q. SBI additionally enables a detailed analysis of parameter-wise posterior uncertainty, presented in Appendix R. Across cells, amplitude-based features were matched most reliably, whereas timing- and

width-related features exhibited larger discrepancies, consistent with forward-model misspecification in real tissue.

*Table 1.* Multi-electrode stimulation prediction accuracy averaged across 37 parasol RGCs. Approximately 200 thousand simultaneous stimulus injections were delivered at either a 3-electrode triangle or line on the MEA at either the axon or the soma of the RGC (Appendix O). Accuracy, balanced accuracy, false positive rate (FPR), and false negative rate (FNR) are shown for biophysical models (gradient descent and SBI), a data-driven MLP baseline trained directly on multi-electrode responses from held-in cells, and prior heuristic baselines. $\uparrow$ indicates higher is better.

| Method | Acc. $\uparrow$ | Bal. Acc. $\uparrow$ | FPR | FNR |
|---|---|---|---|---|
| HH (GD) | **0.904** | **0.893** | 0.077 | 0.137 |
| HH (SBI) | 0.873 | 0.841 | 0.072 | 0.245 |
| MLP | 0.859 | 0.833 | 0.107 | 0.228 |
| Independent | 0.854 | 0.812 | 0.074 | 0.302 |
| Superposition | 0.843 | 0.811 | 0.102 | 0.276 |

**HH RGC model predictions of simultaneous multi-electrode stimulation.** To evaluate generalization beyond single-electrode supervision, we predicted spike/no-spike responses to previously unseen simultaneous multi-electrode stimuli using models fit to each cell's EI features and single-electrode thresholds. Multi-electrode response maps were analyzed for 37 parasol cells spanning 7 retinal preparations, with currents swept simultaneously over fixed electrode triplets (Appendix T). Because the biophysical forward model $f_{\mathrm{stim}}$ is deterministic, experimental responses were binarized by labeling a stimulus as "spike" if its empirical spike probability exceeded $0.5$ across repeats. Prior to evaluating multi-electrode predictions, a per-electrode current scale factor was estimated for each cell by matching the model's predicted single-electrode threshold to the experimentally measured threshold, accounting for unmeasured

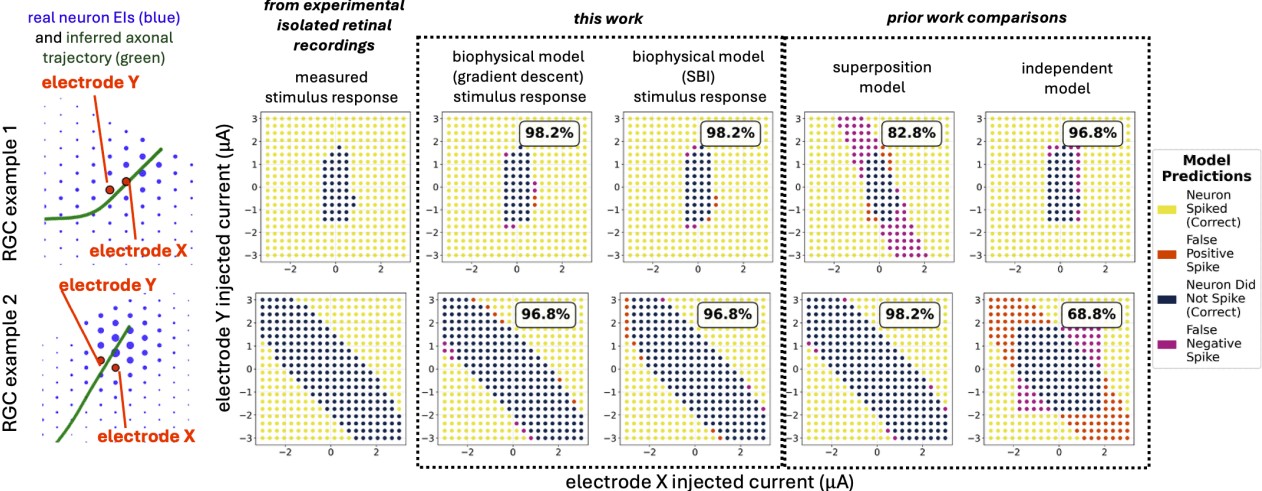

*Figure 5.* **Prediction of simultaneous multi-electrode stimulation responses in real RGCs. Left:** For each example cell, the recorded EI (blue) and inferred axonal trajectory (green) are shown with the stimulated electrodes (X,Y) highlighted. **Middle:** Experimentally measured response maps obtained by sweeping currents on electrodes X and Y; a stimulus is labeled "spike" if empirical spike probability exceeds 0.5 across repeats. Although stimulation was delivered simultaneously at three electrodes, maps are shown as two-dimensional cross-sections with the third electrode held at zero current for clarity (see Appendix T for full 3-electrode plots). **Right:** Predicted response maps from HH models fit from EI features and single-electrode thresholds (gradient-based point estimate; SBI), compared to two prior baselines (Vasireddy et al., 2026; Vilkhu et al., 2025). The top row shows a stereotyped *independent-site* case, where electrodes lie at separated locations along the inferred trajectory and the response is well-approximated by a rectangular decision region. The bottom row shows a stereotyped *superposition-like* case, where electrodes lie near the same putative axonal site and the response is approximately linear in normalized currents. Across both regimes, the fitted HH model captures the observed interaction structure without committing to either limiting assumption.

shunting of injected current. This scale factor is derived entirely from the same single-electrode threshold measurements used during fitting and does not introduce additional supervision (Appendix T).

Figure 5 illustrates two representative maps demonstrating neural response to simultaneous stimulation that emphasize distinct interaction geometries. In the top example, the stimulated electrodes lie at well-separated locations along the inferred axonal trajectory, and the measured response is well-approximated by an independent-site structure. In the bottom example, the stimulated electrodes lie near the same putative axonal site, and the measured response exhibits an approximately linear (superposition-like) boundary. Predictions from HH models fit by gradient-based point estimation and by SBI are shown alongside two prior state of the art baselines reflecting these limiting assumptions of models proposed in Vasireddy et al. (2026); Vilkhu et al. (2025): the *independent-site* baseline predicts spiking when either electrode exceeds its single-electrode threshold, while the *superposition* baseline sums normalized currents (relative to thresholds) to produce an approximately linear boundary. Across both examples, the fitted HH model captures the appropriate interaction regime without selecting it a priori.

**Population-level comparison.** To quantify performance at scale, predictions were aggregated across all tested cur-

rent combinations across all 37 cells. Table 1 reports binary prediction accuracy for HH models and baseline methods, showing that HH models fit from minutes of single-electrode data achieved the highest accuracy on unseen multi-electrode stimulation patterns. A detailed breakdown of prediction accuracy across cells and electrode configurations, including the dependence of performance on electrode placement relative to the inferred axonal morphology, is provided in Appendix T.

To assess whether the performance gain of our approach stems from the biophysical inductive bias rather than model expressiveness, we evaluated a multilayer perceptron (MLP; 128→64→1, ReLU, dropout) baseline. Unlike the HH model, which requires only single-electrode recordings during fitting, the MLP requires multi-electrode stimulation responses for training. We therefore trained it on responses from 27 cells and evaluated on 7 held-out cells, repeating this procedure across 20 random cell-level splits to avoid evaluating on an atypically easy or difficult subset. The MLP received as input the same EI-derived features used to fit the HH model: sodium, potassium, and capacitive peak amplitudes and pairwise sodium peak timing differences at the stimulating electrodes, together with stimulus amplitudes normalized by single-electrode thresholds. The MLP achieved high accuracy on the held-in training dataset of cells (Appendix T). However, despite having direct access to

multi-electrode stimulation responses from training cells, a substantial information advantage the HH model never has, the MLP achieves only 0.859 accuracy and 0.833 balanced accuracy on held-out cells, underperforming HH (GD) on every metric. These results are consistent with the biophysical inductive bias playing an important role in generalizing to new cells from single-electrode data, though we note that more expressive or carefully tuned data-driven models may narrow this gap with sufficient multi-electrode training data.

## 5. Related Work

A substantial body of work has developed Bayesian and probabilistic methods for estimating parameters of biophysical neuron models from noisy intracellular recordings. Early contributions introduced efficient optimization- and filtering-based approaches for fitting HH-type models to voltage- and current-clamp data, including linear-regression-based optimization methods (Huys et al., 2006) as well as particle filtering and Kalman-filter-based techniques for joint inference over latent states and model parameters (Huys & Paninski, 2009; Lankarany et al., 2013). More recently, hybrid approaches combining neural networks with Hodgkin–Huxley models have emerged. In particular, simulation-based inference (SBI) methods enable likelihood-free Bayesian inference for complex mechanistic neuron models using neural density estimators and amortized inference (Beck et al., 2022; Bernaerts et al., 2025), while other approaches leverage adversarial networks (Kim et al., 2025) or evolutionary optimization techniques such as differential evolution (Naudin et al., 2022). Collectively, these works demonstrate that principled inference of detailed biophysical models is feasible when informative intracellular measurements are available.

Recent work has begun to explore fitting biophysical neuron models using extracellular recordings (Tanoh et al., 2026), but existing approaches are largely restricted to simulated data. Consequently, there is currently no established method for inferring detailed Hodgkin-Huxley-type neuron models from real extracellular recordings, where membrane voltages are unobserved. In this work, we address this gap and apply our approach to enrich existing methods for determining retinal ganglion cell multi-electrode stimulation thresholds (Vasireddy et al., 2026). Because a calibrated biophysical model can predict responses to arbitrary stimulus combinations without additional measurement, it enables exploration of a much larger space of candidate current patterns than is feasible through empirical testing alone, a capability that is directly relevant to ongoing efforts seeking high-fidelity reproduction of naturalistic activity patterns using targeted electrical stimulation (Phillips et al., 2026; Lotlikar et al., 2023; Shah et al., 2024).

## 6. Discussion

We presented a novel framework for learning biophysically detailed multi-compartment HH models directly from high-density extracellular recordings, enabling cell-specific modeling at population scale without intracellular access. The biophysical inductive bias enables generalization to unseen stimulus combinations that purely data-driven approaches cannot replicate without direct supervision, suggesting a path toward model-based exploration of stimulus spaces too large to characterize empirically (Vasireddy et al., 2026; Phillips et al., 2026).

**Relationship between EI feature fit quality and multi-electrode prediction.** As shown in Figure 4, forward model mismatch in real tissue produces imperfect EI feature recovery, with spike duration exhibiting the largest discrepancies. Despite this, multi-electrode prediction accuracy remains high, because stimulation responses are primarily determined by axonal orientation and location relative to the stimulating electrodes (Vilkhu et al., 2025), information encoded most directly in the peak amplitude features. Timing features are less sensitive to axonal geometry and more sensitive to ion channel kinetics and membrane capacitance, and their misspecification therefore has limited bearing on the downstream prediction task. Accurate inference of these parameters would become essential for predicting responses to stimulus waveforms beyond those tested here, such as width-modulated stimuli, where membrane temporal dynamics play a larger role.

**Gradient descent versus simulation-based inference.** GD and SBI exhibit complementary strengths that reflect the same forward model mismatch. In simulated settings with injected current noise, SBI outperforms GD because it explicitly models stochastic variability that GD ignores. On real data, GD outperforms SBI because SBI is more sensitive to distributional shift between simulated training data and real observations. Performance differences may also reflect the large space of design decisions in both methods, from feature weighting in the GD loss to summary statistic choice for SBI, making exhaustive comparison across implementations impractical.

**Limitations and future directions.** The primary forward model limitation is the assumption of a homogeneous extracellular medium, which contributes to feature misspecification and likely accounts for a portion of residual prediction errors. Further analysis of per-cell prediction accuracy and failure modes is provided in Appendix T. Broader directions for future work include extending the framework to synaptically coupled networks of neurons and enabling closed-loop settings in which inferred models adaptively guide precise stimulus selection.

## Impact Statement

This paper presents a framework for learning biophysical models of neurons from extracellular recordings, with a focus on predicting responses to electrical stimulation in isolated ex vivo macaque retina. The immediate translational goal is to reduce the empirical calibration burden for epiretinal prostheses, which currently requires hours of patient testing that could be replaced by minutes of model-based prediction. All experiments in this work were conducted on isolated retinal tissue obtained from terminally anesthetized macaques euthanized during research performed by other laboratories, in accordance with IACUC approval and national guidelines. No experiments were performed on live animals or humans.

Several important limitations bound the current scope of this work. Our models are validated on responses to $150\ \mu s$ charge-balanced triphasic pulses delivered to up to 3 simultaneous electrodes; generalization to other stimulus waveforms or larger electrode configurations remains to be demonstrated. Translation to chronic in vivo implants may additionally require addressing electrode-tissue interface drift, which limits the timescale over which a calibrated biophysical model remains valid in a way that depends on implant stability. The homogeneous extracellular medium assumption in the forward model is a further limitation; ongoing work to directly measure heterogeneous retinal resistivity may improve forward model fidelity in future iterations. More broadly, the ability to predict neural responses to arbitrary stimulation patterns from a biophysical model raises longer-term questions about the responsible use of such tools as brain-computer interface technology matures. We believe these considerations are best addressed as the field advances toward clinical deployment and encourage ongoing community engagement on this topic.

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

# Appendices

## A. Author Contributions

A. Lotlikar initiated and led the project, developed the JAXLEY forward models for straight-axon and RGC simulations, conceived the electrical image feature extraction approach, designed the EI-based location initialization procedure, set up the real-data processing pipeline, implemented and evaluated the gradient-based optimization framework, and led the manuscript writing. I.C. Tanoh led the model misspecification and simulation-based inference components, performed the corresponding experiments, and wrote the SBI sections of the manuscript. P. Vasireddy provided technical guidance on multi-electrode stimulation and led multi-electrode stimulation data collection together with R. Vilkhu and A. Lotlikar in the Chichilnisky Lab. A. Lanpouthakoun performed the ablation study, contributed to development of the JAXLEY forward models, and contributed substantially to manuscript writing and editing. R. Vilkhu provided technical guidance and NEURON code for retinal ganglion cell modeling on which the JAXLEY forward models were based. M. Sommeling and A.J. Phillips contributed experimental resources and assisted with with data collection. A. Sher and A. Litke developed the 512-electrode system used for RGC stimulation and recording. S.W. Linderman, E.J. Chichilnisky, and S. Mitra jointly supervised the project in equal capacity, providing scientific and technical guidance and manuscript editing.

## B. Axial Conductance Definition

Axial conductance between electrically connected compartments $i$ and $j$ is defined as

$$G_{ij}^{\text{ax}} = \left[ \frac{1}{2} \rho_{\text{ax}} \left( \frac{l_i}{\pi r_i^2} + \frac{l_j}{\pi r_j^2} \right) \right]^{-1}, \tag{5}$$

and $G_{ij}^{\text{ax}} = 0$ otherwise, where $\rho_{\text{ax}}$ is the axial resistivity.

## C. Computation of Extracellular Stimulation Responses

Here we define the forward model $f_{\text{stim}}$ and describe the computation of extracellular stimulation response.

To simulate neuronal responses to extracellular electrical stimulation, we incorporate electrode-induced extracellular potentials into the Hodgkin–Huxley membrane dynamics. Each stimulating electrode is modeled as a point source injecting current into a homogeneous conductive half-space. The extracellular potential induced by electrode $m$ at spatial location $\mathbf{x}$ is given by

$$V_m(\mathbf{x}, t) = \frac{I_m(t)}{2\pi\sigma \left\| \mathbf{x} - \mathbf{x}_m \right\|}, \tag{6}$$

where $I_m(t)$ is the applied electrode current, $\sigma$ is the tissue conductivity, and $\mathbf{x}_m$ is the electrode location.

For membrane segment $i$ located at $\mathbf{x}_i$, the induced extracellular potential is $V_{m,i}(t) = V_m(\mathbf{x}_i, t)$. The effect of extracellular stimulation on the membrane dynamics is incorporated by defining the stimulus current term in the Hodgkin–Huxley equations as

$$I_i^{\text{stim}}(t) = \sum_{j \sim i} \sum_m G_{ij}^{\text{ax}} \big( V_{m,j}(t) - V_{m,i}(t) \big), \tag{7}$$

where the sum over $j \sim i$ denotes electrically connected neighboring membrane segments and $G_{ij}^{\text{ax}}$ is the axial conductance between segments $i$ and $j$. This construction reflects the fact that extracellular electrical stimulation primarily affects neural membrane voltage by inducing axial currents in the neural cytoplasm as opposed to directly inducing transmembrane currents.

Given a stimulation waveform $\{I_m(t)\}_m$, we simulate the membrane response by inserting $I_i^{\text{stim}}(t)$ into Equations 1–3 with initial conditions:

$$V_i(0) = -70 \, \text{mV}, \quad m_i(0) = 0.2, \quad h_i(0) = 0.2, \quad n_i(0) = 0.2. \tag{8}$$

**Spike detection.** Let $t_0 = \min\{t : I_i^{\text{stim}}(t) \geq 0\}$. We consider the set $\{t : t \geq t_0\}$ to be the simulation window. We define a model-predicted action potential to have occurred if two conditions are met: (1) the membrane potential of any compartment exceeds a threshold of 0 mV, and (2) this suprathreshold activity propagates to neighboring compartments. The binary outcome of this detection determines the value of $f_{\text{stim}}(\boldsymbol{\theta}, I) \in \{0, 1\}$.

Figure 6 illustrates this process for a cable cell subject to triphasic extracellular stimulation at varying intensities near threshold.

**Simulated Membrane Voltage Response to Extracellular Triphasic Stimulation**
**At Varying Intensities Centered Around Stimulus Threshold**

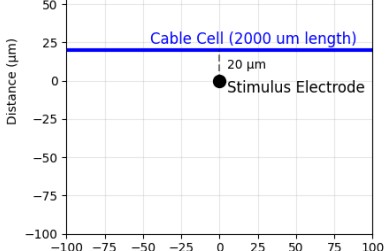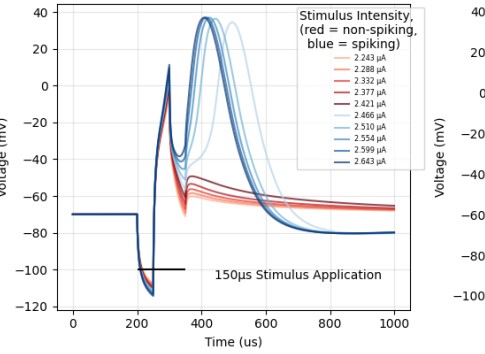

*Figure 6.* Visualizing neural membrane voltage response to extracellular stimulation. **Left:** Electrode and cable geometry, with the stimulus electrode positioned 20 $\mu$m from a 2000 $\mu$m cable. **Center:** Membrane potential at the compartment closest to the electrode (20 $\mu$m away) for stimulus amplitudes spanning the threshold. Subthreshold responses (red) show passive depolarization followed by return to rest, while suprathreshold responses (blue) elicit full action potentials. **Right:** Membrane potential at a compartment 200 $\mu$m from the electrode, demonstrating action potential propagation for suprathreshold stimuli. The black bar indicates the 150 $\mu$s stimulus duration.

## D. Differential Spike Probability for Stimulation with One Electrode

Here we define $P_{\mathrm{stim},m}(\boldsymbol{\theta}, I)$, a differentiable implementation to predict the spiking response of a neuron with parameters $\theta$ to extracellular single-electrode stimulation at electrode $m$ with intensity $I$. We apply this function to train compartment models with gradient descent based on their stimulus thresholds. In this work, we only simulate and model charge-balanced triphasic current pulse consistent with the pulses delivered in our experimental data collection protocol. The baseline waveform $w(t)$ consists of three phases, each of duration $\tau = 50$ $\mu$s:

$$
w(t) = \begin{cases} -\frac{2}{3} & t \in [t_0, t_0 + \tau) \\ +1 & t \in [t_0 + \tau, t_0 + 2\tau) \\ -\frac{1}{3} & t \in [t_0 + 2\tau, t_0 + 3\tau) \\ 0 & \text{otherwise} \end{cases} \tag{9}
$$

where $t_0$ denotes the stimulus onset time. Note that the phases sum to zero, ensuring charge balance. The applied electrode current for electrode $m$ at stimulus amplitude $I$ is then $I_m(t) = I \cdot w(t)$. Our implementation of $P_{\mathrm{stim},m}(\boldsymbol{\theta}, I)$ assumes that the pulse is triphasic charge-balanced 150$\mu$s. However, alternative waveforms (e.g., biphasic pulses, longer phase durations) could be substituted by modifying $w(t)$ and recalibrating the detection parameters below.

**Spike detection.** Let $t_{\mathrm{ind}} = t_0 + 170$ $\mu$s denote the indicator time, comprising the 150 $\mu$s stimulus duration plus a 20 $\mu$s buffer for action potential initiation. We define the *critical voltage* as the maximum membrane potential across all compartments at the indicator time:

$$
V_{\mathrm{crit}}(\boldsymbol{\theta}, m, I) = \max_i V_i(t_{\mathrm{ind}}; \boldsymbol{\theta}, m, I), \tag{10}
$$

where $V_i(t; \boldsymbol{\theta}, m, I)$ is the membrane potential of compartment $i$ at time $t$, given biophysical parameters $\boldsymbol{\theta}$, stimulating electrode $m$, and stimulus amplitude $I$. For waveforms with different temporal structure, $t_{\mathrm{ind}||}$ should be adjusted to ensure sufficient time for spike initiation while remaining sensitive to threshold crossing.

**Differentiable probability surrogate.** The binary spike detector is non-differentiable with respect to the biophysical parameters. To enable gradient-based optimization, we define a smooth surrogate probability function:

$$P_{\text{stim},m}(\boldsymbol{\theta}, I) = \sigma\left(\frac{V_{\text{crit}}(\boldsymbol{\theta}, m, I) - V_{\text{thresh}}}{\gamma(V_{\text{crit}})}\right), \tag{11}$$

where $\sigma(x) = (1 + e^{-x})^{-1}$ is the logistic sigmoid and $V_{\text{thresh}}$ is the spike threshold. We compute $V_{\text{thresh}}$ in practice by sweeping $I$ over a range encompassing the threshold stimulus current and setting $V_{thresh}$ to be the value which maps cases where $f_{stim} = 1$ probabilities greater than 0.5 and $f_{stim} = 0$ to probabilities less than 0.5. For the critical voltage at the example indicator time visually depicted in Figure6 middle panel, $V_{thresh} \approx -45\text{mV}$ separates injection currents that do not stimulate an action potential from those that do. This in effect sets stimulation with the threshold current to 0.5. The scale parameter $\gamma$ controls the sharpness of the transition and is defined asymmetrically around threshold:

$$\gamma(V_{\text{crit}}) = \begin{cases} \gamma_+ & V_{\text{crit}} > V_{\text{thresh}} \\ \gamma_- & V_{\text{crit}} \leq V_{\text{thresh}} \end{cases} \tag{12}$$

with $\gamma_+ = 10 \text{ mV}$ and $\gamma_- = 5 \text{ mV}$. The asymmetric scaling accounts for the observation that subthreshold responses decay more rapidly than suprathreshold responses saturate. This sigmoid-based surrogate is applicable to any stimulus waveform; only the scale parameters $\gamma_\pm$ may require recalibration for substantially different pulse shapes or durations.

Figure 6 illustrates the membrane voltage response to triphasic stimulation at varying intensities near threshold, demonstrating the transition from subthreshold passive responses to suprathreshold action potential generation and propagation.

## E. Electrical Image Feature Extraction

To compare simulated and experimentally recorded extracellular waveforms for parameter inference, we extract a set of physically interpretable features from the electrical image (EI). These features capture key aspects of spike shape and propagation and enable reliable gradient-based optimization.

Let $\Phi_m(t) \in \mathbb{R}$ denote the extracellular potential recorded at electrode $m$ as a function of time. Retinal ganglion cell EIs exhibit three canonical spike components: an early positive *capacitive* peak, a dominant negative *sodium* influx peak, and a late positive *potassium* outflux peak. Feature extraction proceeds by isolating these components and computing both amplitude- and timing-based descriptors.

**Peak amplitudes.** For each electrode $m$, we first identify the sodium peak time

$$t_m^{\text{Na}} = \arg\min_t \Phi_m(t).$$

Using this reference, we define component-specific masked traces:

$$\Phi_m^{\text{cap}}(t) = \mathbf{1}[t \leq t_m^{\text{Na}}]\, \Phi_m(t), \quad \Phi_m^{\text{K}}(t) = \mathbf{1}[t \geq t_m^{\text{Na}}]\, \Phi_m(t),$$

$$\Phi_m^{\text{Na}}(t) = -\mathbf{1}[t_m^{\text{cap}} \leq t \leq t_m^{\text{K}}]\, \Phi_m(t),$$

where

$$t_m^{\text{cap}} = \arg\max_t \Phi_m^{\text{cap}}(t), \qquad t_m^{\text{K}} = \arg\max_t \Phi_m^{\text{K}}(t)$$

are the capacitive and potassium peak times, respectively. The per-electrode peak amplitudes are then defined as

$$A_m^{\text{cap}} = \max_t \Phi_m^{\text{cap}}(t), \quad A_m^{\text{Na}} = \max_t \Phi_m^{\text{Na}}(t), \quad A_m^{\text{K}} = \max_t \Phi_m^{\text{K}}(t).$$

**Timing features.** We define two classes of timing features: action potential duration and propagation delay. Action potential duration at electrode $m$ is defined as the time between the capacitive and potassium peaks, while propagation delay is defined as the difference in sodium peak arrival times between electrode pairs.

To make these timing features differentiable with respect to waveform samples, we approximate peak times using a soft-argmax operator. For each component trace $\Phi_m^{(c)}(t)$, with $c \in \{\text{cap}, \text{Na}, \text{K}\}$, the soft-argmax peak time is given by

$$\hat{t}_m^{(c)} = \sum_t t\, \text{softmax}_t\big(\Phi_m^{(c)}(t)\big).$$

Using these differentiable peak times, we define

$$\Delta t_m^{\text{dur}} = \hat{t}_m^{\text{K}} - \hat{t}_m^{\text{cap}}, \qquad \Delta t_{m_1,m_2}^{\text{prop}} = \hat{t}_{m_1}^{\text{Na}} - \hat{t}_{m_2}^{\text{Na}}.$$

These time-difference features are invariant to global temporal shifts in the electrical image and therefore avoid explicit waveform alignment between simulated and experimentally recorded extracellular potentials.

## F. Loss Function with Extracellular Features for MSE Gradient Based Inference

This appendix provides the full definition of the differentiable loss function $\mathcal{L}(\boldsymbol{\theta})$ used for gradient-based biophysical parameter inference.

Let $f_{\text{EI}}(\boldsymbol{\theta})$ denote the electrical image produced by the forward model for parameter vector $\boldsymbol{\theta}$. From this simulated EI, we extract the following features using the procedures described in Appendix E:

- $A_{m,\text{sim}}^{(k)}(\boldsymbol{\theta})$: the peak amplitude of component $k \in \{\text{cap}, \text{Na}, \text{K}\}$ at electrode $m$,

- $\Delta t_{m,\text{sim}}^{\text{dur}}(\boldsymbol{\theta})$: the action potential duration at electrode $m$,

- $\Delta t_{m_1,m_2,\text{sim}}^{\text{prop}}(\boldsymbol{\theta})$: the propagation delay between electrodes $m_1$ and $m_2$.

**Observed features.** Let $f_{\text{EI}}^{\text{obs}}$ denote the experimentally recorded electrical image. The corresponding observed features $\left(A_{m,\text{obs}}^{(k)}, \Delta t_{m,\text{obs}}^{\text{dur}} \Delta t_{m_1,m_2,\text{obs}}^{\text{prop}}\right)$ are extracted using identical procedures applied to the experimental recording.

**Stimulation thresholds.** Let $(T_m^+, T_m^-)$ denote the experimentally measured positive and negative stimulus thresholds for electrode $m$, defined as the stimulus amplitudes at which the neuron spikes with $50\%$ probability. Let $P_{\text{stim},m}(\boldsymbol{\theta}, I)$ denote the differentiable surrogate for the stimulation response probability at electrode $m$ and stimulus amplitude $I$ (Appendix D).

**Regularization.** The term $\mathcal{R}(\boldsymbol{\theta})$ denotes a regularization penalty on the parameters, weighted by $\lambda \geq 0$.

**Full loss function.** The full loss function is defined as

$$
\begin{aligned}
\mathcal{L}(\boldsymbol{\theta}) = &\sum_{k \in \{\text{Na}, K, \text{cap}\}} w_k \sum_m \left(A_{m,\text{sim}}^{(k)}(\boldsymbol{\theta}) - A_{m,\text{obs}}^{(k)}\right)^2 \\
&+ w_{\text{prop}} \sum_{m_1 < m_2} \left(\Delta t_{m_1,m_2,\text{sim}}^{\text{prop}}(\boldsymbol{\theta}) - \Delta t_{m_1,m_2,\text{obs}}^{\text{prop}}\right)^2 \\
&+ w_{\text{dur}} \sum_m \left(\Delta t_{m,\text{sim}}^{\text{dur}}(\boldsymbol{\theta}) - \Delta t_{m,\text{obs}}^{\text{dur}}\right)^2 \\
&+ w_{\text{stim}} \sum_m \left[\left(P_{\text{stim},m}(\boldsymbol{\theta}, T_m^+) - 0.5\right)^2 + \left(P_{\text{stim},m}(\boldsymbol{\theta}, T_m^-) - 0.5\right)^2\right] \\
&+ \lambda \mathcal{R}(\boldsymbol{\theta}),
\end{aligned}
\tag{13}
$$

All EI timing features are implemented using soft argmin/max operators, and the stimulation surrogate $P_{\text{stim},m}$ is differentiable by construction, making $\mathcal{L}(\boldsymbol{\theta})$ fully differentiable with respect to $\boldsymbol{\theta}$. We investigated several methods to balance contributions from heterogeneous feature groups, such as adapting all feature weights $\{w_k, w_{\text{prop}}, w_{\text{stim}}\}$ with GradNorm (Chen et al., 2017) to equalize gradient magnitudes across loss terms. We settled upon employing loss balancing across feature groups. See code for implementation. Gradients are computed using automatic differentiation, and parameters are optimized with Adam using bounded reparameterizations to enforce physiological constraints on conductances, radii, and spatial locations.

## G. Waveform-Based EI Loss with Variable Duration

To provide a baseline for EI fitting, a waveform-level loss was implemented that compares simulated and observed electrical images (EIs) directly in the time domain. This baseline was constructed to handle the practical setting in which EIs can

have different durations: spike sorters typically return only the non-zero portion of the spike waveform, and simulated EIs may exhibit different spike widths across biophysical parameters. As a result, pointwise waveform differences are not well-defined without an explicit alignment procedure.

Let $\Phi^{(1)} \in \mathbb{R}^{T_1 \times M}$ and $\Phi^{(2)} \in \mathbb{R}^{T_2 \times M}$ denote two EIs (time $\times$ electrodes), with a shared number of electrodes $M$ but potentially different durations $T_1 \neq T_2$. A shift-aligned mean-squared error (MSE) was defined by zero-padding one EI and evaluating the MSE over all discrete temporal offsets. Concretely, $\Phi^{(1)}$ was padded along the time dimension with $T_2 - 1$ zeros on both sides,

$$\tilde{\Phi}^{(1)} = \mathrm{pad}(\Phi^{(1)}, T_2 - 1) \in \mathbb{R}^{(T_1 + 2(T_2 - 1)) \times M},$$

and for each offset $k \in \{0, \ldots, T_1 + T_2 - 2\}$ an aligned sub-window of length $T_2$ was extracted,

$$\Phi_k^{(1)} = \tilde{\Phi}^{(1)}[k : k + T_2] \in \mathbb{R}^{T_2 \times M}.$$

The MSE for offset $k$ was then

$$\mathrm{MSE}(k) = \frac{1}{T_2 M} \left\| \Phi_k^{(1)} - \Phi^{(2)} \right\|_F^2,$$

and the waveform loss was defined as the minimum over alignments,

$$\mathcal{L}_{\mathrm{wave}}(\Phi^{(1)}, \Phi^{(2)}) = \min_{k \in \{0, \ldots, T_1 + T_2 - 2\}} \mathrm{MSE}(k). \tag{14}$$

This shift-and-pad construction enabled waveform-level comparison of EIs with mismatched durations without resampling or landmark detection.

**Motivation for feature-based losses.** Although Eq. (14) is an intuitive similarity measure, it induces an objective that is poorly behaved for gradient-based biophysical inference. In particular, the minimizing offset is piecewise constant as a function of the underlying biophysical parameters, and residual misalignment or duration mismatch can dominate pointwise errors even when the underlying spike shapes are physiologically similar. Consequently, gradients are often driven by phase and noise rather than by biophysically meaningful variations (Appendix G). These considerations motivated the feature-based objective in Appendix F, where physically interpretable descriptors (peak amplitudes and shift-invariant timing differences; Appendix E) are matched instead of raw waveforms, yielding a smoother landscape for optimization and more reliable parameter recovery.

# H. Simulation-Based Inference Overview

Simulation-based inference (SBI) provides a likelihood-free framework for inferring model parameters in settings where the likelihood is intractable or unavailable, but forward simulation is feasible (Cranmer et al., 2020; Deistler et al., 2025a; Tejero-Cantero et al., 2020). This is the case for many biophysical neuron models, where complex nonlinear dynamics, high-dimensional latent states, and measurement noise make explicit likelihood evaluation impractical.

Let $\theta \in \Theta$ denote the model parameters of interest, and let $x \in \mathcal{X}$ denote the observation vector used for inference. In SBI, one assumes access to a simulator that generates synthetic observations $x \sim p(x \mid \theta)$ for any chosen $\theta$, without requiring an explicit form for the likelihood. Given a prior distribution $p(\theta)$, the goal is to approximate the posterior distribution

$$p(\theta \mid x_{\mathrm{obs}}) \propto p(x_{\mathrm{obs}} \mid \theta) \, p(\theta), \tag{15}$$

conditioned on an observed dataset or feature vector $x_{\mathrm{obs}}$.

Modern SBI methods leverage neural density estimators to approximate either the posterior directly, the likelihood, or the likelihood-to-evidence ratio. In this work, we focus on neural posterior estimation (NPE), in which a conditional density estimator $q_\phi(\theta \mid x)$, parameterized by neural network weights $\phi$, is trained on simulated pairs $(\theta_i, x_i)$ generated by sampling $\theta_i \sim p(\theta)$ and simulating $x_i \sim p(x \mid \theta_i)$. The parameters $\phi$ are optimized by maximizing the conditional log-likelihood of the training samples,

$$\max_{\phi} \mathbb{E}_{p(\theta)p(x|\theta)} \left[ \log q_\phi(\theta \mid x) \right]. \tag{16}$$

In our experiments, the conditioning variable $x$ is not the full voltage trace, but the same low-dimensional observation vector used for gradient-based fitting: EI features extracted from the simulated extracellular response together with stimulus

thresholds. Thus, the inference network takes these summary features as input and outputs a conditional posterior density over the unknown parameters. We parameterize $q_\phi(\theta \mid x)$ as an invertible normalizing-flow density estimator conditioned on $x$. This parameterization provides a flexible posterior approximation that can capture non-Gaussian and potentially multimodal uncertainty, rather than restricting the posterior to a single Gaussian distribution. In the stochastic-current experiments, the target parameter vector includes both the biophysical parameters $\boldsymbol{\theta}$ and the auxiliary noise parameter $\sigma_I$, corresponding to the standard deviation of the injected-current noise.

Once trained, the learned density estimator can be evaluated at the observed feature vector $x_{\text{obs}}$ to yield an explicit approximation of the posterior distribution over parameters. This posterior can be used to compute summary statistics such as posterior means, variances, or credible intervals, or to obtain point estimates such as the maximum a posteriori (MAP) estimate.

A key advantage of SBI is that it naturally captures parameter uncertainty and identifiability through the shape and spread of the inferred posterior, rather than relying solely on point estimates. SBI methods are also amortized: once trained, the same inference network can be reused to infer parameters from new observations generated under the same model class, enabling efficient inference across multiple datasets or experimental conditions.

## I. Straight-axon Simulation setup

We model a single unbranched axon as a straight cylindrical cable of total length $L = 2000 \ \mu$m, discretized into compartments of length $\Delta x = 2 \ \mu$m, yielding $N = 1000$ compartments. All compartments share a common radius and homogeneous Hodgkin–Huxley sodium and potassium conductance densities. The kinetics of these channels were chosen as in Fohlmeister et al. (2010) (Table 5).

**Action potential initiation for electrical image simulation.** To generate electrical images, we simulate a single action potential propagating along the axon. At $t = 0$, the membrane voltage is initialized with the first 50 compartments (corresponding to the proximal 100 $\mu$m) set to 0 mV, while all remaining compartments are set to the resting potential of $-70$ mV. This suprathreshold initialization at one end of the axon triggers an action potential that propagates distally according to the Hodgkin–Huxley dynamics without any additional injected current. The resulting transmembrane currents are then used to compute the extracellular potential at each electrode via Equation (4), yielding the simulated electrical image.

**Electrode geometry.** We use a local patch of $M = 7$ electrodes in a hexagonal configuration with spacing $s = 30 \ \mu$m. Six electrodes lie on a ring in the $yz$-plane centered at the origin, with coordinates $(0, s\cos(2\pi k/6), s\sin(2\pi k/6))$ for $k = 0, \ldots, 5$, and one electrode at the center $(0, 0, 0)$.

**Axon geometry.** In the straight-axon experiments, axon geometry is specified by three orientation angles $(\theta, \phi, \psi)$ and a perpendicular offset distance $d$ from the electrode patch. These parameters are used only to define the geometry of a straight cable via two endpoints. Specifically, $(d, \theta, \phi, \psi)$ deterministically define endpoints $\mathbf{p}_-$ and $\mathbf{p}_+$ separated by the fixed axon length $L$.

This endpoint representation matches the control-point parameterization used in the full RGC model. Given $\mathbf{p}_\pm$, compartment centers are obtained by linearly interpolating $N+1$ evenly spaced points along the segment and taking midpoints of adjacent points, enforcing fixed length $L$ and uniform spacing $\Delta x$.

**Parameter learning.** We learn the axon radius $r$, sodium and potassium conductance densities $(g_{\text{Na}}, g_{\text{K}})$, and the axon geometry represented by the endpoint pair $(\mathbf{p}_-, \mathbf{p}_+)$. All other biophysical and tissue parameters are fixed.

For our gradient-based approach, initialization uses the angle-distance parameterization $(d, \theta, \phi, \psi)$, with all scalar parameters set to the midpoint of their allowed ranges (Table 2). Endpoints $\mathbf{p}_\pm$ are then constructed deterministically from these values using the straight-axon geometry mapping.

For simulation-based inference experiments, axon geometry is parameterized directly using $(d, \theta, \phi, \psi)$, with independent uniform priors over all parameters within the bounds listed in Table 2.

*Table 2.* Parameter bounds for straight-axon identifiability experiments.

| Parameter | Min | Max |
|---|---|---|
| Axon offset distance $d$ ($\mu$m) | 10 | 30 |
| Orientation angle $\theta$ (rad) | $-\pi/4$ | $\pi/4$ |
| Orientation angle $\phi$ (rad) | $\pi/4$ | $3\pi/4$ |
| Spin angle $\psi$ (rad) | $-\pi/4$ | $\pi/4$ |
| Axon radius $r$ ($\mu$m) | 1 | 5 |
| Sodium conductance $g_{\text{Na}}$ (S/cm$^2$) | 0.1 | 0.3 |
| Potassium conductance $g_{\text{K}}$ (S/cm$^2$) | 0.1 | 0.3 |

## J. Ablation Study for Inference of Straight Axon Biophysical Parameters with Gradient Descent

We evaluate the contribution of each feature group to biophysical parameter recovery using the simulated straight-axon dataset described in Section 4. We decompose the loss function (Equation 13) into four feature groups: peak amplitudes (A), action potential duration (D), propagation velocity (V), and stimulation probability (E). For each ablation condition, we set the weights of excluded feature groups to zero (i.e. $w_k$ for $k \in \{\text{Na}, \text{K}, \text{cap}\}$, $w_{\text{prop}}, w_{\text{dur}}, w_{\text{stim}}$). We performed inference across approximately 50 randomly sampled axon configurations and evaluated parameter recovery using the coefficient of determination ($R^2$) between inferred and ground-truth values.

**Results without stimulation features.** Figure 7 shows parameter recovery for all combinations of the three EI-derived feature groups (D, A, V). Peak amplitudes alone (A) achieved strong recovery across all parameters: axon radius ($R^2 = 0.991$), Na$^+$ conductance ($R^2 = 0.993$), K$^+$ conductance ($R^2 = 0.987$), and distance to electrode ($R^2 = 0.937$). In contrast, duration alone (D) and propagation velocity alone (V) showed substantially weaker performance, particularly for conductance densities (D: $R^2 = 0.732$ for $g_{\text{Na}}$; V: $R^2 = 0.153$ for $g_{\text{Na}}$) and axon radius (D: $R^2 = 0.511$; V: $R^2 = 0.505$).

Combining amplitude with either timing feature (D+A or A+V) yielded near-optimal performance, with marginal improvement from the full combination (D+A+V: $R^2 = 0.997$ for radius, $R^2 = 0.995$ for $g_{\text{Na}}$, $R^2 = 0.994$ for $g_{\text{K}}$). Notably, combining duration and velocity without amplitude (D+V) produced intermediate results, indicating that timing features alone provide partial but incomplete constraints on biophysical parameters.

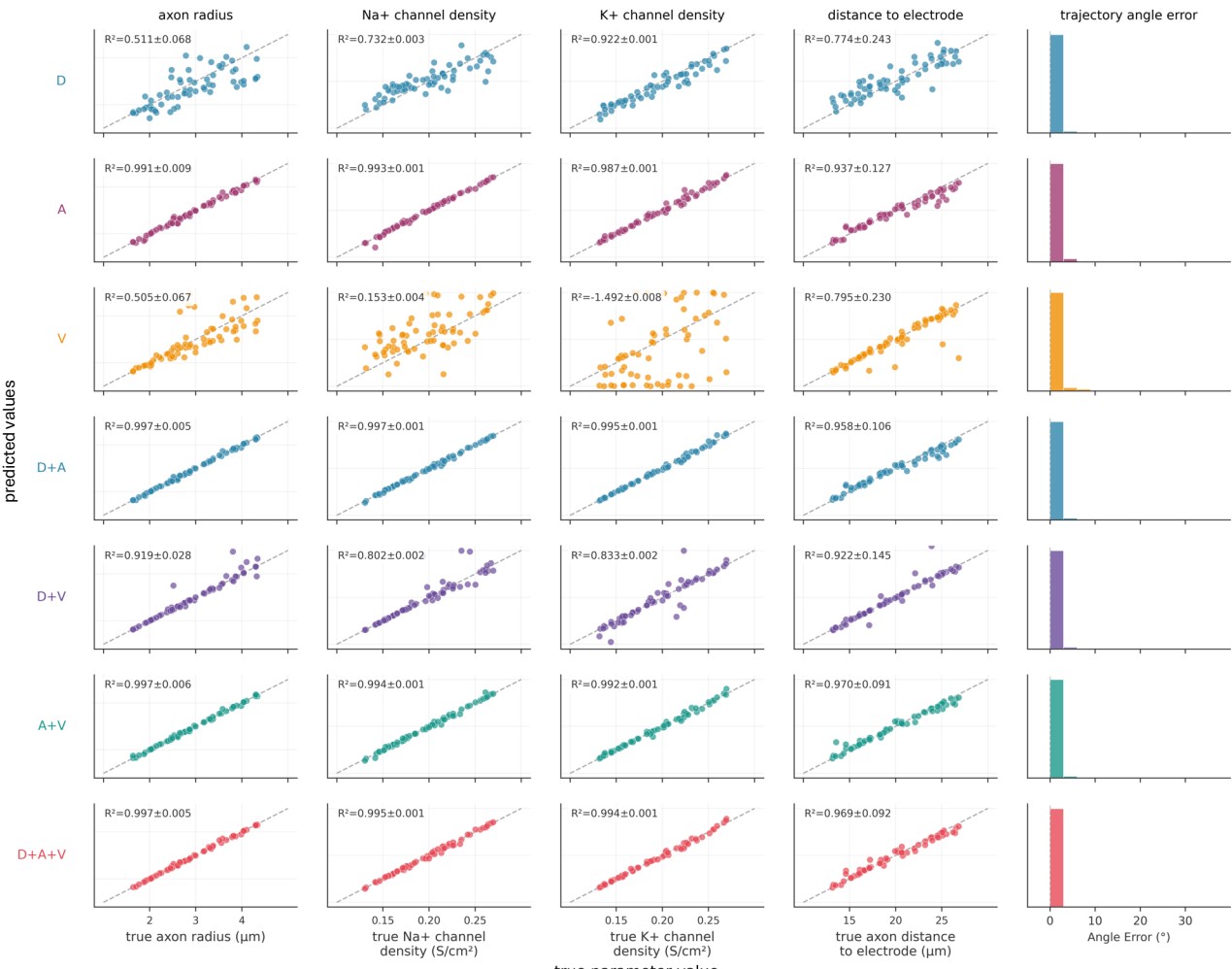

*Figure 7.* Ablation study: parameter recovery using EI-derived features only. Each row corresponds to a different combination of loss features: duration (D), amplitude (A), velocity (V), and their combinations. Columns show recovery of axon radius, Na$^+$ conductance density, K$^+$ conductance density, and axon-to-electrode distance, with $R^2$ values annotated. The rightmost column shows the distribution of trajectory angle errors. Amplitude features alone (A) achieve strong recovery across all parameters ($R^2 > 0.93$), while duration (D) and velocity (V) alone show weaker performance. Combining all three EI features (D+A+V) yields near-perfect recovery. Dashed lines indicate equality between inferred and ground-truth values. Values reported after $\pm$ denote the standard error of residuals ($\sigma_{\mathrm{residual}}/\sqrt{n}$), shown as an indicator of fit dispersion rather than a sampling standard error of $R^2$ itself.

**Results with stimulation features.** Figure 8 shows the effect of incorporating stimulation probability (E) into the loss function. Stimulation features alone performed poorly, yielding negative $R^2$ values for radius ($-2.247$), Na$^+$ conductance ($-1.861$), and K$^+$ conductance ($-3.705$), indicating predictions worse than a constant baseline. This reflects the limited geometric information available from threshold measurements at a single electrode.

However, stimulation features provided complementary information when combined with EI-derived features. Adding E to the amplitude-only condition (A+E) maintained strong performance while the full model (D+A+V+E) achieved the best overall recovery: $R^2 = 0.973$ for radius, $R^2 = 0.993$ for $g_{\mathrm{Na}}$, $R^2 = 0.995$ for $g_{\mathrm{K}}$, $R^2 = 0.997$ for electrode distance, and median trajectory angle error below $5$.

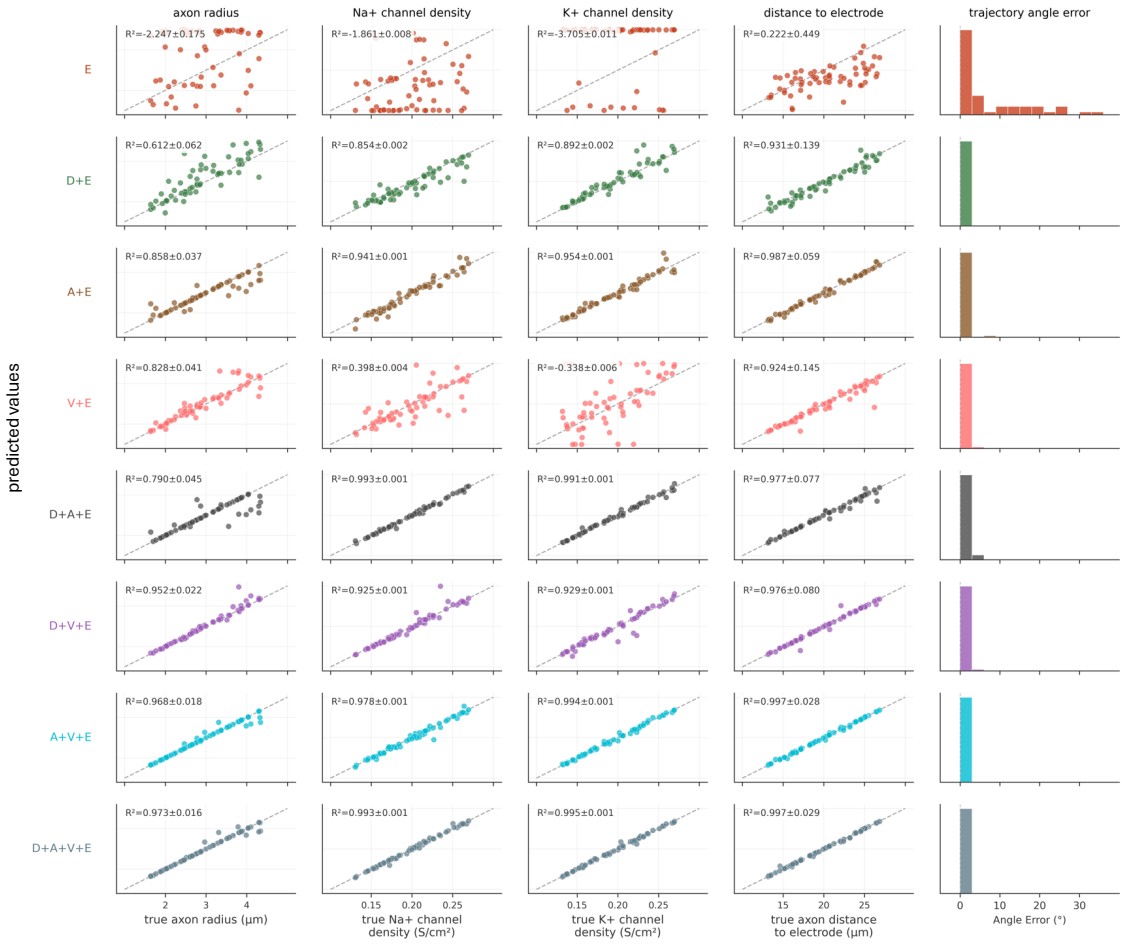

*Figure 8.* Ablation study: parameter recovery incorporating extracellular stimulation features (E). Each row shows a different combination of loss features including stimulation thresholds. Stimulation features alone (E, top row) yield poor recovery with negative $R^2$ values, reflecting insufficient geometric constraints. However, adding stimulation features to EI-derived features improves performance: the full model (D+A+V+E, bottom row) achieves the best overall recovery ($R^2 > 0.97$ for distance, $R^2 > 0.99$ for conductances) with trajectory angle errors consistently below 10. Format follows Figure 7.

**Interpretation.**    As summarized in Table 3, peak amplitudes emerge as the dominant feature for parameter inference, achieving $R^2 > 0.93$ across all parameters when used alone—performance that other individual features cannot match. This is likely due to the fact that this feature directly encode both geometric information (signal magnitude scales with electrode-axon distance) and conductance densities (which determine ionic current magnitudes). Timing features provide complementary geometric constraints—propagation velocity is sensitive to axon orientation, while duration reflects the interplay between capacitance and conductance—but cannot independently constrain conductance parameters. Stimulation thresholds, while uninformative in isolation, may help resolve degeneracies between radius and conductance when combined with amplitude information, as threshold depends on the spatial profile of the activating function along the axon.

*Table 3.* Summary of ablation results ($R^2$, mean $\pm$ std). Bold indicates best single-feature performance; underline indicates best overall.

| Features | Radius | $g_{\mathrm{Na}}$ | $g_{\mathrm{K}}$ | Distance |
|---|---|---|---|---|
| *Without stimulation* | | | | |
| D | $0.511 \pm 0.068$ | $0.732 \pm 0.003$ | $0.922 \pm 0.001$ | $0.774 \pm 0.243$ |
| A | $\mathbf{0.991 \pm 0.009}$ | $\mathbf{0.993 \pm 0.001}$ | $\mathbf{0.987 \pm 0.001}$ | $\mathbf{0.937 \pm 0.127}$ |
| V | $0.505 \pm 0.067$ | $0.153 \pm 0.004$ | $-1.492 \pm 0.008$ | $0.795 \pm 0.230$ |
| D+A+V | $0.997 \pm 0.005$ | $0.995 \pm 0.001$ | $0.994 \pm 0.001$ | $0.969 \pm 0.092$ |
| *With stimulation* | | | | |
| E | $-2.247 \pm 0.175$ | $-1.861 \pm 0.008$ | $-3.705 \pm 0.011$ | $0.222 \pm 0.449$ |
| D+A+V+E | $\underline{0.973 \pm 0.016}$ | $\underline{0.993 \pm 0.001}$ | $\underline{0.995 \pm 0.001}$ | $\underline{0.997 \pm 0.029}$ |

## K. Analysis of posterior uncertainty for SBI approach on Straight-Axon model

In this section, we report additional experimental results for simulation-based inference (SBI) in the straight-axon setting (Section 4.1). For each parameter of interest, SBI produces a posterior distribution conditioned on the observed EI features. As discussed in the main text, when using the posterior mean as a point estimator, SBI increasingly outperforms the gradient-based approach as the level of injected current noise increases (Figure 9).

We focus here on the behavior of the posterior uncertainty. As expected, the posterior standard deviation increases monotonically with the noise level $\sigma_I$ (Figure 10A). Intuitively, higher noise in the injected current leads to increased variability in the resulting membrane voltages and, consequently, in the extracted EI features. This variability spreads posterior probability mass over a larger region of parameter space, making precise localization of the true parameters from a single observation more challenging. Nevertheless, even at elevated noise levels, the posterior standard deviation remains small relative to the corresponding ground-truth parameter values, indicating that uncertainty grows in a controlled manner with noise.

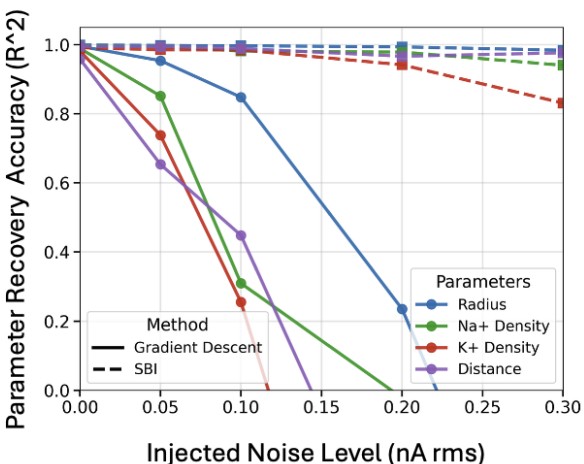

*Figure 9.* Performance vs noise level for SBI and gradient-based method in the straight axon case

Finally, we assess posterior calibration by examining coverage. Across all noise levels, the ground-truth parameters consistently lie within high-probability regions of the inferred posterior, as quantified by the proportion of simulations in which the true parameters fall within $\pm 2$ posterior standard deviations of the posterior mean (Figure 10B). This indicates that SBI remains well-calibrated in the straight-axon regime, even under substantial observation noise.

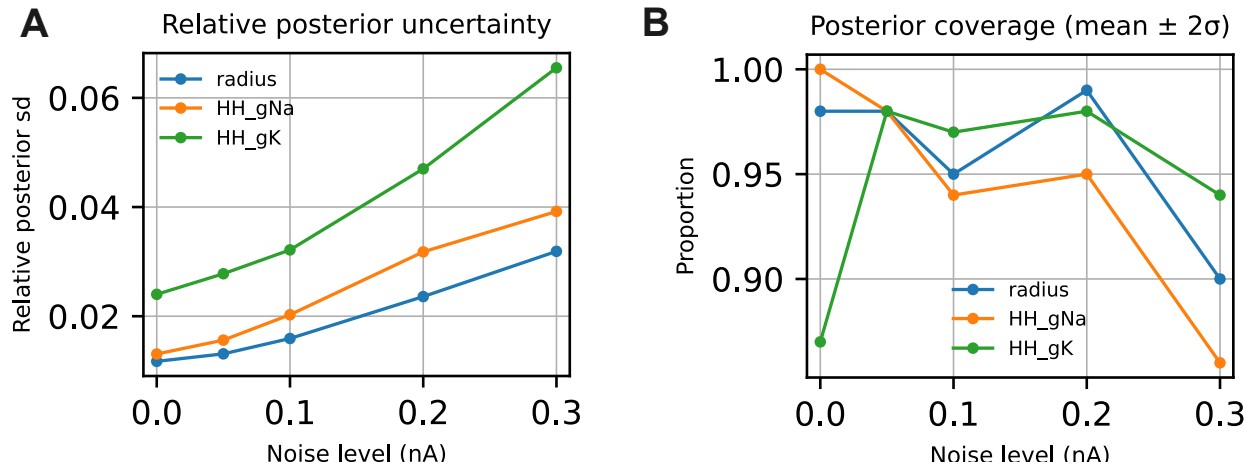

*Figure 10.* **Posterior uncertainty and calibration for the straight axon model. (A)** Relative posterior uncertainty as a function of observation noise. For each noise level, 100 ground-truth parameter configurations are simulated and posterior distributions are inferred using simulation-based inference (SBI). The reported values correspond to the posterior standard deviation normalized by the ground-truth parameter value, averaged across simulations. **(B)** Posterior coverage as a function of observation noise. We report the proportion of simulations for which the ground-truth parameter lies within $\pm 2$ posterior standard deviations of the posterior mean.

## L. Retinal Ganglion Cell Biophysical Modeling

**Multi-compartment Hodgkin–Huxley model.** Each RGC was represented as a collection of connected cylindrical compartments. We adopted a four-ion channel model (Na$^+$, K$^+$, Ca$^{2+}$, and KCa) for mammalian RGC membrane voltage dynamics. Ion channel kinetics were measured via intracellular patch-clamp recordings in previous work (Fohlmeister et al., 2010). In this model, the transmembrane time-varying voltage $V$ of each compartment $i$ obeys the following differential equation:

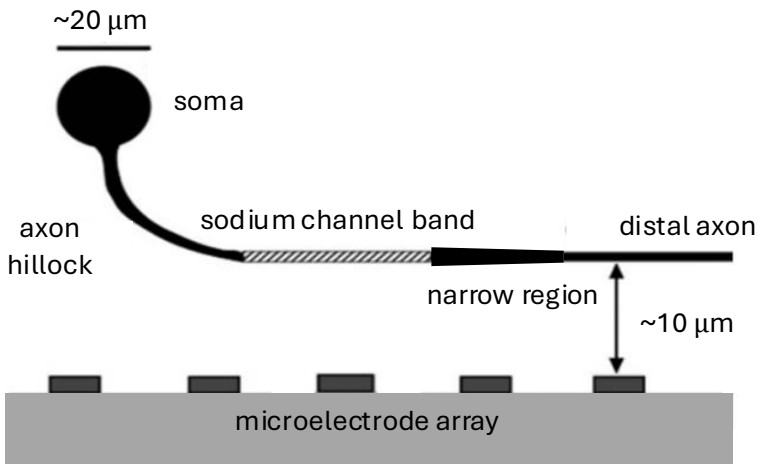

*Figure 11.* Regions of RGC HH model

$$S_i C_m \frac{dV_i}{dt} = I_i^{\text{stim}} - S_i J_i^{\text{ion}} + \sum_{j \sim i} G_{ij}^{\text{ax}} (V_j - V_i). \tag{17}$$

with

$$
\begin{aligned}
J_i^{\text{ion}} =& \bar{g}_{\text{Na},i}\, m_i^3 h_i \left(V_i - E_{\text{Na}}\right) + \bar{g}_{\text{K},i}\, n_i^4 \left(V_i - E_{\text{K}}\right) \\
& + \bar{g}_{\text{Ca},i}\, c_i^3 \left(V_i - E_{\text{Ca}}\right) + \bar{g}_{\text{K,Ca},i} \left(V_i - E_{\text{K}}\right) + g_{\text{pas},i} \left(V_i - E_{\text{pas}}\right).
\end{aligned}
\tag{18}
$$

where $I_i^{\text{stim}}$ is a stimulus current applied to compartment $i$, $S_i$ is its membrane surface area, $\bar{g}_{\text{Na},i}, \bar{g}_{\text{K},i}, \bar{g}_{\text{Ca},i}, \bar{g}_{\text{K,Ca},i}, g_{\text{pas},i}$ are its maximal ion channel and leak conductances(S/cm$^2$), $m_i, h_i, n_i, c_i \in [0,1]$ are its time-varying gating states, $E_{\text{Na}}, E_{\text{K}}, E_{\text{Ca}}, E_{\text{pas}}$ are ionic Nernst potentials (Table 4), and $G_{ij}^{\text{ax}}$ is the axial conductance coupling a compartment $i$ to its neighbor $j$. The gating variables follow first-order kinetics of the form $dx/dt = \alpha_x(V_i)(1-x) - \beta_x(V_i)x$ for $x \in \{m_i, h_i, n_i, c_i\}$, with voltage-dependent rate functions $\alpha_x(V), \beta_x(V)$ measured by Fohlmeister et al. (2010) at 35°C (the experimental temperature) in Table 5.

**Action potential initiation for electrical image simulation.** To generate electrical images for RGCs, we simulate a single action potential initiated at the soma. At $t = 0$, the membrane voltage of all compartments is initialized to the passive resting potential ($-65$ mV), except for soma compartments which are set to a depolarized voltage ($-50$ mV). This suprathreshold initialization at the soma triggers an action potential that propagates through the axon hillock and along the distal axon according to the Hodgkin–Huxley dynamics without any additional injected current. The resulting transmembrane currents are then used to compute the extracellular potential at each electrode via Equation (4), yielding the simulated electrical image.

**Model regions and parameters.** RGC model compartments were grouped into five distinct regions: dendrites, soma, axon hillock, sodium channel band, and distal axon (Figure 11). Table 6 summarizes the bounds of maximal ion conductances used in fitted models for each region, and Table 7 provides the corresponding bounds on compartment length and radius. These parameter bounds were informed by previous modeling studies characterizing variability in retinal ganglion cell morphology (Vilkhu et al., 2025; Kish et al., 2023). Importantly, Nernst potentials (Table 4) and ion channel kinetics (Table 5) were taken to be measured constants, while sodium and potassium channel densities and neural morphology relative to electrodes were allowed to vary across fitted RGCs.

*Table 4.* Nernst potentials for ion channels (mV).

|  | $E_{\text{Na}}$ | $E_{\text{K}}$ | $E_{\text{Ca}}$ | $E_{\text{pas}}$ |
|---|---|---|---|---|
| Value | 60.60 | $-101.34$ | $\dfrac{RT}{2F}\ln\left(\dfrac{[Ca^{2+}]_e}{[Ca^{2+}]_i(t)}\right)$ | $-64.58$ |

*Table 5.* Rate constants for voltage-gated ion channels. Voltage $V$ in mV.

| Sodium (Na$^+$) | Potassium (K$^+$) | Calcium (Ca$^{2+}$) |
|---|---|---|
| $\alpha_m = \dfrac{-2.725(V+35)}{e^{-0.1(V+35)} - 1}$ | $\alpha_n = \dfrac{-0.09575(V+37)}{e^{-0.1(V+37)} - 1}$ | $\alpha_c = \dfrac{-1.362(V+13)}{e^{-0.1(V+13)} - 1}$ |
| $\beta_m = 90.83\,e^{-(V+60)/20}$ | $\beta_n = 1.915\,e^{-(V+47)/80}$ | $\beta_c = 45.41\,e^{-(V+38)/18}$ |
| $\alpha_h = 1.817\,e^{-(V+52)/20}$ | | |
| $\beta_h = \dfrac{27.25}{1 + e^{-0.1(V+22)}}$ | | |

*Table 6.* Baseline values of maximal conductance (S/cm$^2$) parameters, by region (Kish et al., 2023).

| Region | $g_{\text{Na}}$ | $g_{\text{K}}$ | $g_{\text{Ca}}$ | $g_{\text{K,Ca}}$ | $g_{\text{pas}}$ |
|---|---|---|---|---|---|
| Soma | 0.06 | 0.035 | 0.00075 | 0.00017 | 0.0001 |
| Axon hillock | 0.15 | 0.09 | 0.00075 | 0.00017 | 0.0001 |
| Sodium channel band | 0.42 | 0.25 | 0.00075 | 0.00011 | 0.0001 |
| Narrow region | 0.15 | 0.09 | 0.00075 | 0.00017 | 0.0001 |
| Distal axon | 0.10 | 0.05 | 0.00075 | 0.00020 | 0.0001 |

*Table 7.* Baseline values of compartment size parameters, by region (Kish et al., 2023).

| Region | Length (μm) | Radius (μm) |
|---|---|---|
| Soma | 20 | 10.0 |
| Axon hillock (AH) | 40 | 1.5 |
| Sodium channel band (SOCB) | 40 | linear taper from AH to NR |
| Narrow region (NR) | 80 | 0.4 |
| Distal axon | 360 | 0.5 |

## M. Simulated Experiments with Retinal Ganglion Cells

To validate biophysical parameter inference for the heterogeneous RGC model described in Section 4 and Appendix L, we performed simulated recovery experiments analogous to the straight-axon sweep (Figure 3).

**Parameter sampling.** Ground-truth RGC parameters were sampled from physiologically realistic bounds (Table 8). For each simulated cell, sodium and potassium conductance densities were varied for the SOCB and distal axon regions, while conductances in other regions (dendrite, soma, axon hillock, narrow region) were held fixed. Distal axon radius was also varied, with a global radius scale factor applied to all compartments.

**Geometry sampling.** Axon trajectories were parameterized by $N_{\mathrm{cp}} = 8$ control points with spacing $\Delta s = 100$ $\mu$m. For each simulated cell, control point locations were generated by perturbing a baseline straight trajectory defined as a sequence of points along the positive $x$-axis: $(0, 0), (\Delta s, 0), (2\Delta s, 0), \ldots$ Each control point was perturbed by adding independent uniform noise in the range $[-30, 30]$ $\mu$m to both $x$ and $y$ coordinates. The $z$-coordinate (depth) of the first control point (soma) was fixed at $-30$ $\mu$m, and subsequent control points used a fixed axon depth of $-10$ $\mu$m.

*Table 8.* Parameter bounds for simulated RGC recovery experiments. Rows with a single value indicate fixed parameters; ranges indicate varied parameters.

| Region | $g_{\mathrm{Na}}$ (S/cm$^2$) | $g_{\mathrm{K}}$ (S/cm$^2$) | Radius ($\mu$m) |
|---|---|---|---|
| Dendrite | 0.06 | 0.035 | 0.5 |
| Soma | 0.105 | 0.0625 | 10.0 |
| Axon hillock | 0.1875 | 0.1225 | 1.5 |
| SOCB | 0.25–0.75 | 0.15–0.5 | — |
| Narrow region | 0.195 | 0.1225 | 0.4 |
| Distal axon | 0.05–0.25 | 0.025–0.1 | 0.16–1.5 |

**Simulation and feature extraction.** For each sampled parameter configuration, we simulated an action potential using the RGC forward model and computed the electrical image on a subset of the 512-electrode array. Electrodes were filtered to include only those within 120 $\mu$m of the axon trajectory in both $x$ and $y$ dimensions. From each simulated EI, we extracted the same engineered features used for inference: peak amplitudes ($A_m^{\mathrm{cap}}$, $A_m^{\mathrm{Na}}$, $A_m^{\mathrm{K}}$), action potential duration ($\Delta t_m^{\mathrm{dur}}$), and pairwise propagation delays ($\Delta t_{m_1, m_2}^{\mathrm{prop}}$) for electrode pairs within 40 $\mu$m of each other.

**Inference methods.** We evaluated both gradient-based optimization and simulation-based inference (SBI) on the synthetic RGC dataset. For gradient descent, parameters were initialized at the midpoint of their bounds, and control point locations were initialized at known control point centers. Optimization was performed using Adam with feature-specific learning rates, minimizing the loss function defined in Appendix F. For SBI, we trained a neural posterior estimator on 4000 simulated parameter–feature pairs, placing uniform priors over all varied parameters and control point perturbations.

**Results.** Both inference methods accurately recovered ground-truth parameters across $N = 1000$ simulated RGCs. Table 9 summarizes the recovery accuracy for gradient descent and SBI (posterior mean). Conductance densities in the SOCB and distal axon were recovered with, and axon radius recovery achieved. Control point locations were recovered with mean absolute error below. These results demonstrate that the inference framework generalizes from the straight-axon setting to realistic heterogeneous RGC morphologies.

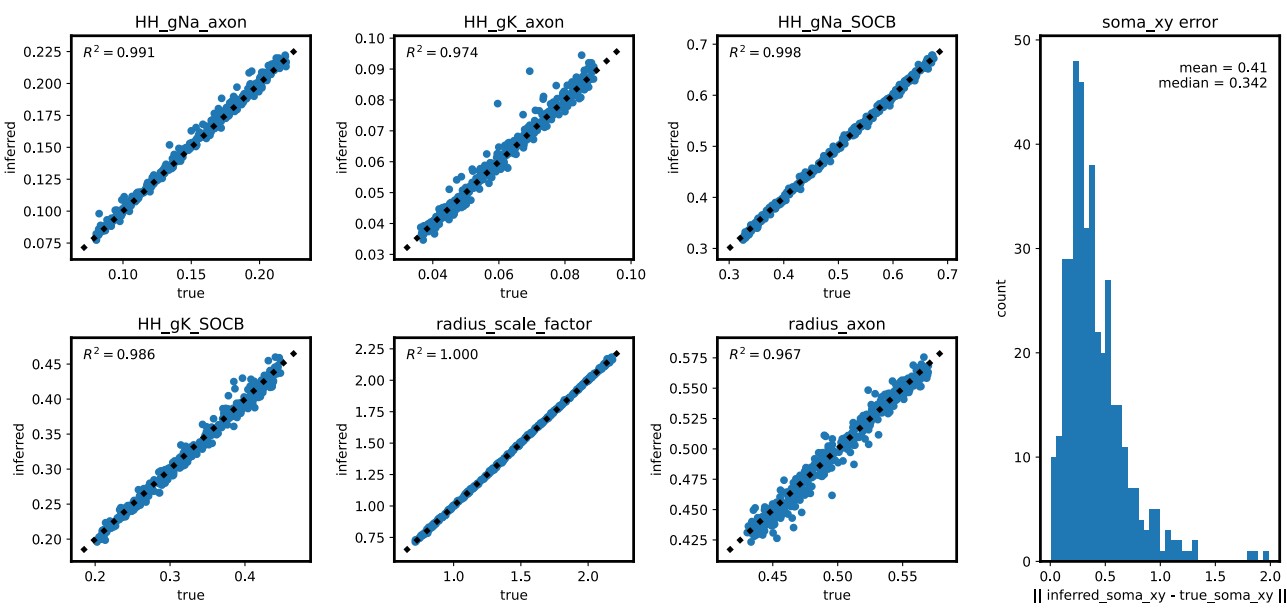

*Figure 12.* Parameter recovery using the simulation-based inferred posterior mean estimate for 400 synthetic RGCs.

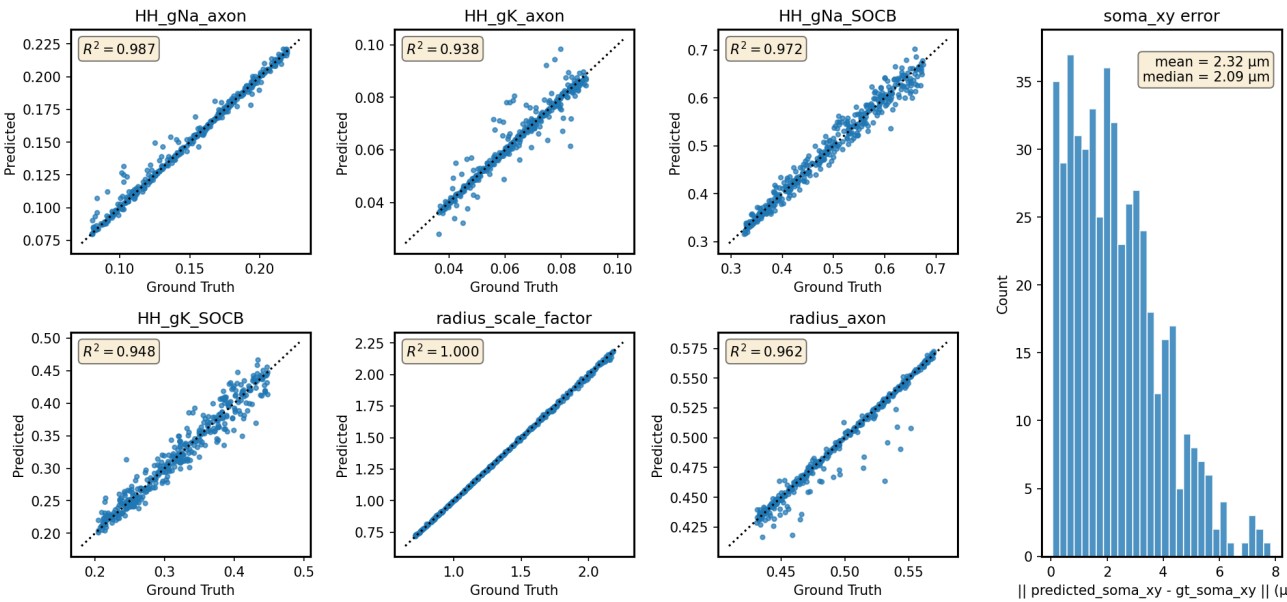

*Figure 13.* Parameter recovery using gradient descent on feature MSE loss for 400 synthetic RGCs.

*Table 9.* Parameter recovery accuracy for simulated RGC experiments ($R^2$, mean across 1000 cells).

| Parameter | Gradient Descent | SBI (Posterior mean) |
|---|---|---|
| $g_{\mathrm{Na,SOCB}}$ | 0.972 | 0.998 |
| $g_{\mathrm{Na,axon}}$ | 0.987 | 0.991 |
| $g_{\mathrm{K,SOCB}}$ | 0.948 | 0.986 |
| $g_{\mathrm{K,axon}}$ | 0.938 | 0.974 |
| $r_{\mathrm{axon}}$ | 0.962 | 0.967 |
| Radius scale factor | 1.000 | 1.000 |

**Comparison to straight-axon results.**    Recovery accuracy for the heterogeneous RGC model was slightly lower than for the straight-axon case (Table 3), reflecting the increased complexity of the inference problem: more parameters, nonlinear geometry, and region-specific conductances. Nevertheless, the feature-based loss function and spline-based geometry parameterization enabled reliable inference across the physiologically realistic parameter space.

## N. Wet Lab Methods for Isolated Retina Experiments

Macaque retinal tissue was obtained from terminally anesthetized rhesus macaques euthanized during the course of research performed by other laboratories, in accordance with Institutional Animal Care and Use Committee (IACUC) approval and national guidelines. No animals were sacrificed specifically for this study.

Immediately following enucleation, eyes were hemisected in room light, the vitreous was removed, and the posterior portion of each eye was stored in darkness in warm (33–35°C), oxygenated, bicarbonate-buffered Ames' solution. Patches of retina approximately 2–3 mm on a side were isolated under infrared illumination and placed ganglion-cell-side down on a custom 512-electrode multielectrode array (MEA) (Hottowy et al., 2012). The MEA consisted of 512 platinum electrodes (diameter 5–15 $\mu$m) arranged in a triangular lattice (Figure 1) with 30 $\mu$m center-to-center pitch, covering a hexagonal area of 0.43 mm$^2$. A transparent dialysis membrane secured the retina against the array during recording. Tissue was continuously superfused with oxygenated Ames' solution at 33–35°C throughout the experiment. A platinum wire encircling the recording chamber served as the distant return electrode. Raw voltage signals from all 512 channels were amplified, bandpass filtered (43–5000 Hz), and digitized at 20 kHz.

**Visual stimulation and cell type identification.**    To identify recorded cell types and extract electrical images, the retina was presented with a dynamic spatiotemporal white noise stimulus (flickering checkerboard) for 15–30 minutes at low photopic light levels. The spike-triggered average (STA) stimulus was computed for each recorded cell and used to classify cell types by clustering on spatial receptive field size and temporal response properties (Chichilnisky, 2001). Analysis focused on ON and OFF parasol retinal ganglion cells (RGCs), two of the four numerically dominant cell types in the primate retina, due to the high signal-to-noise ratio of their recorded spikes. Spikes were identified and sorted using Kilosort2 (Pachitariu et al., 2016). Electrical images (EIs) were computed as the average spatiotemporal voltage waveform across all 512 electrodes during each cell's spike, using spikes identified during visual stimulation.

**Single-electrode stimulation.**    Single-electrode stimulation was delivered through each of the 512 electrodes in sequence while recording from all electrodes simultaneously. Each stimulus consisted of a charge-balanced triphasic current pulse with anodal/cathodal/anodal phases and relative amplitudes 2:−3:1, with 50 $\mu$s per phase (150 $\mu$s total), as described in Appendix D. For each electrode, 25 repeated trials were delivered at each of 40 logarithmically spaced current amplitudes between 0.1 and 4 $\mu$A, for both positive (cathodal-peak) and negative (anodal-peak) polarity pulses. Only negative polarity thresholds were measured for the single-electrode dataset. Stimuli were delivered in pseudorandom order with successive stimulating electrodes spatially separated to ensure a minimum inter-trial interval of approximately 10 ms, allowing neural refractory periods to elapse between successive trials. The full single-electrode sweep across all 512 electrodes required approximately 90 minutes per preparation. Spike probabilities as a function of current amplitude were fit with a sigmoidal activation curve for each electrode-cell pair, yielding an activation threshold defined as the current amplitude at which spiking probability equaled 0.5.

**Three-electrode stimulation.**    For each target cell, three stimulating electrodes were selected based on spike signal-to-noise ratio and spatial position relative to the cell's inferred morphology. Both positive and negative polarity thresholds were measured at the three stimulating electrodes. Simultaneous triphasic stimulation was then delivered at all combinations of current amplitudes swept uniformly over a cubic grid in the three-electrode stimulus space, with 20 or 21 linearly spaced values per electrode (yielding $20^3 = 8,000$ or $21^3 = 9,261$ unique combinations depending on the preparation), with amplitudes ranging from approximately $\pm 1.78$ to $\pm 2.81$ $\mu$A; the exact range and grid size for each cell are recorded in the released dataset (Appendix O). Each combination was intended to be delivered for 20 repeated trials; however, the stimulus computer occasionally crashed mid-collection, resulting in some combinations receiving fewer trials or none at all. The number of trials actually delivered per combination is recorded in the `trials_all` field of the dataset. Stimulus ordering was pseudorandomized with successive electrode groups spatially separated, and a minimum inter-trial interval of 10 ms was enforced between successive stimulations. Electrically evoked spikes were identified using a custom template-matching spike sorter applied to post-stimulus recordings, using EI templates derived from visual stimulation (Vasireddy et al.,

2026). Three-electrode stimulation data collection required approximately 3 hours per cell. The yield of cells with usable three-electrode stimulation data was low relative to the total number of recorded parasol cells, as the selected electrode triplets often did not lie sufficiently close to the cell's axon or soma to reliably elicit spiking across the swept amplitude range.

## O. Real Retinal Ganglion Cell Dataset

This appendix summarizes the real parasol RGC data used in the main text. Experimental protocols are in Appendix N; released data and code are at `https://github.com/amrith1/rgc-ei-fit-hh`.

### O.1. Cohort

All recordings are from isolated *ex vivo* macaque retina on a 512-electrode MEA (30 $\mu$m pitch). We analyzed ON and OFF parasol cells identified from light-evoked receptive fields and Kilosort2 spike sorting.

The released data comprise two sets:

- **Multi-electrode set** (37 cells, 7 retinal preparations): every cell with simultaneous three-electrode stimulation maps, used for multi-electrode prediction evaluation (Table 1, Figure 5).

- **Single-electrode set** (165 cells, 8 preparations): cells with single-electrode threshold measurements used to expand the feature-fitting cohort.

Four cells appear in both sets, giving $37 + 165 - 4 = 198$ unique parasol cells for EI feature recovery (Figure 4) and HH/SBI fitting. There is no population-level train/test split for HH inference: each cell is fit from its own EI and single-electrode thresholds; multi-electrode responses are held out for evaluation on the 37 multi-electrode cells.

### O.2. Data available per cell

**Single-electrode set** (165 **cells**). Each cell has (i) an electrical image (EI) from $\sim$15–30 min of white-noise visual stimulation, and (ii) single-electrode stimulation thresholds measured across the array (full protocol in Appendix N). For the large single-electrode cohort, only negative-polarity thresholds were used as supervision during fitting.

**Multi-electrode set** (37 **cells**). In addition to the EI and single-electrode thresholds, each cell has a full response map to simultaneous stimulation on three selected electrodes (near the soma or axon). Currents were swept on a cubic grid with either 20 or 21 linearly spaced amplitudes per electrode (8,000 or 9,261 unique three-electrode combinations per cell), spanning approximately $\pm 2.81$ $\mu$A. Each combination was targeted for 20 repeated trials ($\sim$19 on average; occasional stimulus interruptions reduced counts for some combinations). Across all 37 cells this yields $\sim 3.2 \times 10^5$ unique stimulus combinations and $\sim 6 \times 10^6$ delivered pulses pooled. For evaluation, a combination is labeled a spike if empirical spike probability exceeds 0.5; model predictions use the same threshold.

### O.3. Collection time (approximate)

Single-electrode thresholds for all 512 electrodes in a preparation required $\sim$90 min (25 repeats $\times$ 40 amplitudes per electrode, 0.1–4 $\mu$A, 150 $\mu$s triphasic pulses). Three-electrode mapping for one cell required $\sim$3 h.

Per-cell breakdowns of multi-electrode prediction accuracy and comparisons across inference methods are in Appendices Q and T.

## P. Fitting Real RGCs: Initial Guess of Location

Gradient-based optimization of RGC biophysical parameters requires a reasonable initialization of compartment locations to avoid local minima and ensure convergence. Here, we define an automated procedure that constructs an initial axon trajectory estimate directly from the electrical image (EI), leveraging the distinct spatiotemporal signatures of different cellular compartments (Figure 14).

**Electrode classification.**    The initialization algorithm first classifies electrodes according to the waveform morphology recorded at each site (Wu et al., 2025). Following established conventions, we distinguish electrodes dominated by somatic, axonal, or dendritic signals based on the relative magnitudes of the three canonical spike components defined in Appendix E. Specifically, for each electrode $m$, we use the capacitive, sodium, and potassium peak amplitudes $A_m^{\text{cap}}$, $A_m^{\text{Na}}$, and $A_m^{\text{K}}$. If the sodium peak amplitude is smaller than the larger of the two positive peaks, i.e., $A_m^{\text{Na}} < \max(A_m^{\text{cap}}, A_m^{\text{K}})$, the electrode is classified as dendritic. Otherwise, the ratio $r_m = A_m^{\text{cap}}/A_m^{\text{K}}$ determines classification: $r_m > 1.6$ indicates an axonal signature, $r_m < 0.8$ indicates a somatic signature, and intermediate values are labeled as mixed (treated as axonal for trajectory estimation).

**Landmark electrode identification.**    Three classes of landmark electrodes anchor the initial geometry, which are depicted visually along with the entire initial location guess process in Figure 14:

1. **Soma electrode**: The electrode with maximum sodium peak amplitude,

$$m_{\text{soma}} = \arg\max_m A_m^{\text{Na}},$$

    reflecting the large somatic current sink.

2. **AIS electrode**: Among the ten highest-amplitude electrodes exceeding a minimum threshold ($A_m^{\text{Na}} > 20\ \mu\text{V}$), the electrode with the earliest sodium peak time,

$$m_{\text{AIS}} = \arg\min_{m \in \mathcal{M}_{\text{top}}} t_m^{\text{Na}},$$

    where $\mathcal{M}_{\text{top}}$ denotes the set of qualifying electrodes. This identifies the axon initial segment (AIS), where action potentials typically initiate.

3. **Axon electrodes**: Electrodes classified as axonal that (i) exceed an amplitude threshold ($A_m^{\text{Na}} > 6\ \mu\text{V}$), (ii) have the largest sodium peak amplitude among electrodes sharing the same sodium peak time $t_m^{\text{Na}}$ (to avoid redundant sampling of the same axonal segment), and (iii) lie more than $150\ \mu\text{m}$ from both the soma and AIS electrodes (to ensure they reflect distal axon rather than perisomatic regions).

**Trajectory construction.**    The electrode landmarks define an ordered sequence: soma → AIS → axon electrodes (sorted by sodium peak time $t_m^{\text{Na}}$, reflecting propagation order). Their 2D coordinates on the electrode array provide waypoints for the axon trajectory. To extend the trajectory beyond the last recorded axon electrode and account for distal axon segments that may not produce detectable signals, we append an extrapolated point at distance $d_{\text{overshoot}} = 500\ \mu\text{m}$ along the line connecting the soma to the most distal axon electrode.

**Spline smoothing and resampling.**    The waypoint sequence is smoothed using cubic B-spline interpolation to produce a continuous trajectory. This interpolation serves two purposes: (i) it regularizes the geometry by enforcing smoothness, consistent with the gradual curvature of real RGC axons, and (ii) it enables resampling at uniform arc-length intervals. The smoothed trajectory is resampled to generate control points spaced at $\Delta s = 40\ \mu\text{m}$ intervals along the arc length. These evenly-spaced control points then serve as the initial parameterization for a second B-spline that defines the final compartment locations during optimization. For real recordings, the $z$-coordinate (distance from electrode array) is initialized at a fixed offset based on typical RGC-to-array distances in epiretinal preparations.

## Q. Models Fit to Real RGC Measurements

Here, we include the full model feature recovery for SBI and Adam continuing main section Figure 4.

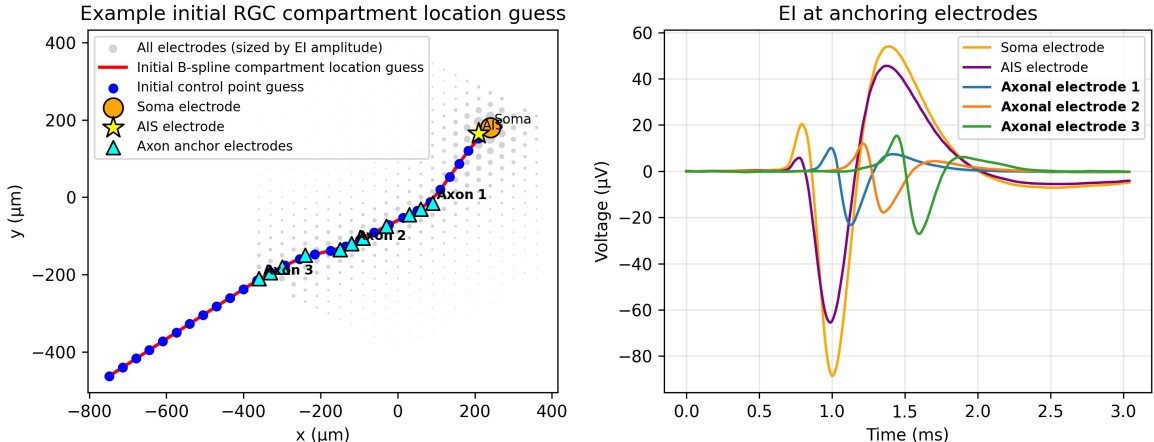

*Figure 14.* Initialization of axon trajectory from electrical image. **Left:** Electrode array with signal amplitudes indicated by dot size. The soma electrode (orange) is identified by maximum sodium peak amplitude. Axon electrodes (blue triangles) are ordered by sodium peak time (red labels) to define the trajectory. **Right:** EI waveforms at selected electrodes along the axon, showing the characteristic triphasic shape and progressive propagation delay from soma to distal axon.

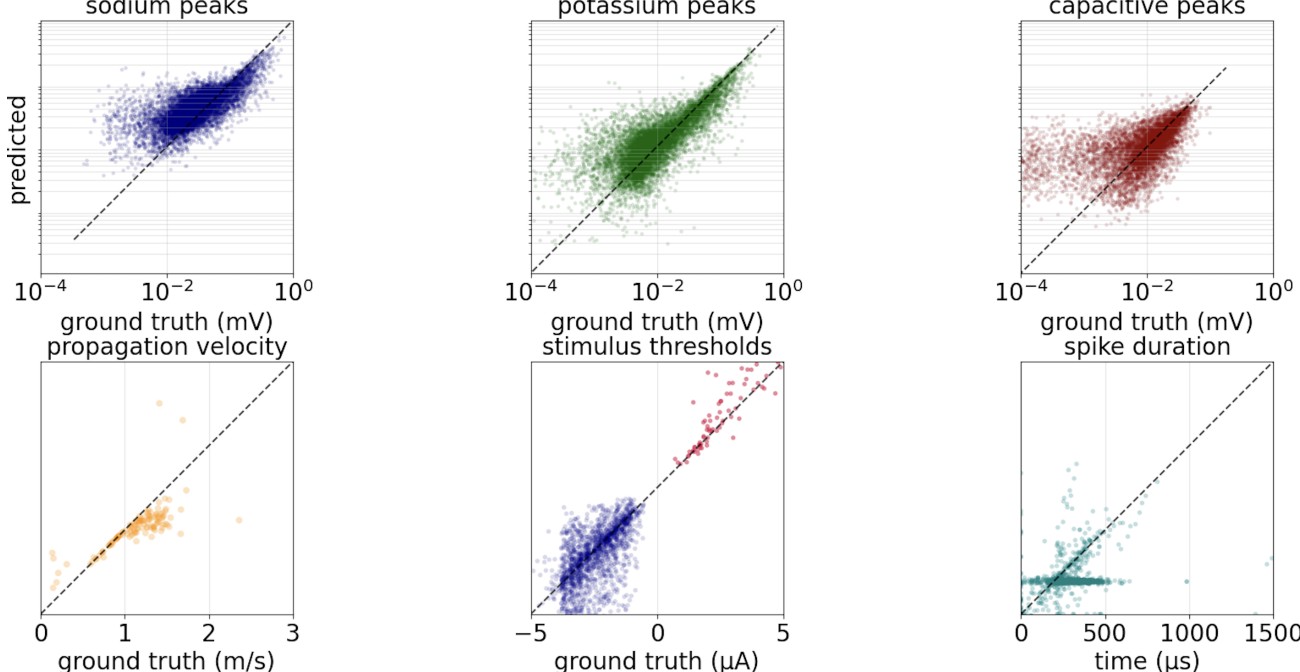

*Figure 15.* Extending figure 4, models trained by gradient on real RGCs were then used to predict the EI features. This plot is identical to that figure with the inclusion of the EI width feature which is misspecified.

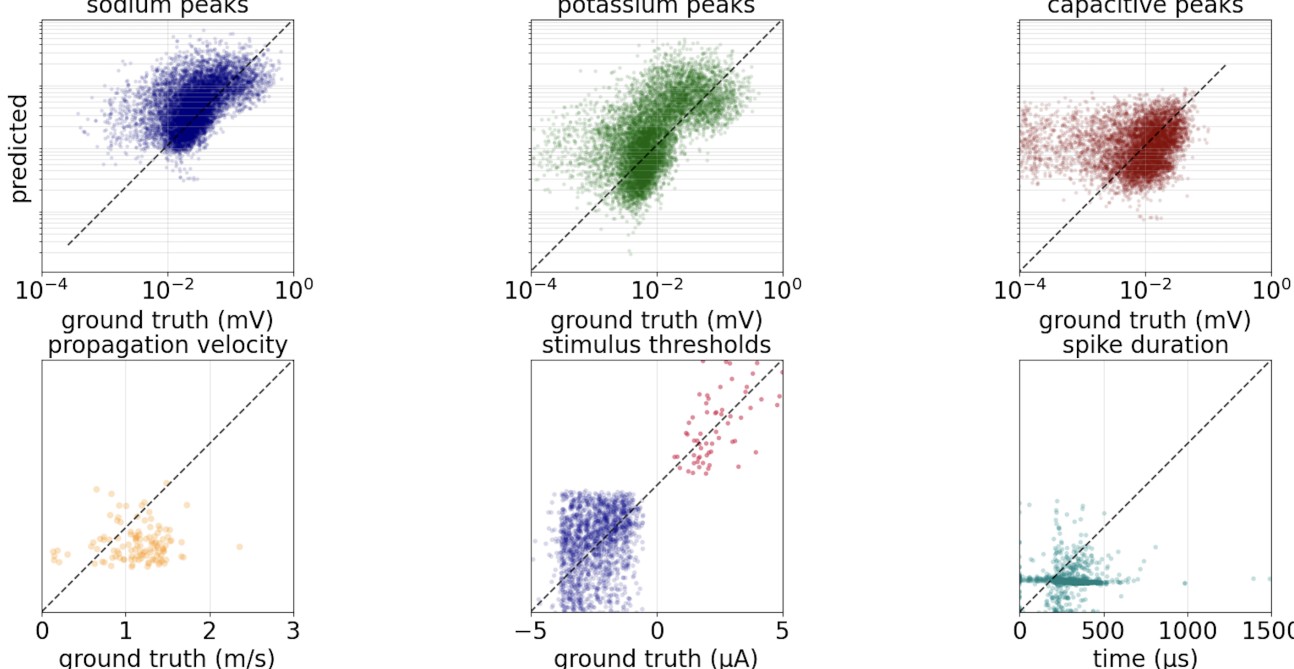

*Figure 16.* Extending figure 4, models trained with SBI on real RGCs were then used to predict the EI features. SBI trained models matched Adam in trends in its capturing of EI features, but its accuracy was degraded.

To quantitatively compare inference accuracy, we report the $R^2$ between predicted and observed features across all real parasol RGCs. Table 10 summarizes these results for models fit via gradient descent and simulation-based inference (SBI). For gradient descent, we report the plain Pearson $R^2$. For SBI, we report the IQR residual down-weighted $R^2$, since SBI produced more extreme outliers than gradient descent, and these large residuals disproportionately degraded the plain $R^2$ despite the presence of visible feature-level trends. Gradient descent achieved higher predictive accuracy on all feature classes, particularly on sodium and potassium amplitudes, while SBI showed degraded performance under model mismatch. Notably, both approaches struggled to capture spike duration features, consistent with misspecification in the forward model.

*Table 10.* Feature prediction performance on real RGCs, reported as $R^2$ between observed and model-predicted features. Gradient descent is reported using plain Pearson $R^2$, while SBI is reported using IQR residual down-weighted $R^2$.

| Method | Sodium | Potassium | Capacitive | Velocity | Duration | Stim. Thresholds |
|---|---|---|---|---|---|---|
| Gradient descent | 0.614 | 0.760 | 0.372 | 0.407 | 0.110 | 0.522 |
| SBI | 0.138 | 0.028 | -1.108 | -1.808 | -0.097 | -0.085 |

## R. Simulation-Based Inference on Real RGC Recordings

In addition to gradient-based point estimation, we applied SBI to infer biophysical parameters for real retinal ganglion cells. Overall, SBI achieved slightly lower predictive accuracy than gradient-based optimization for multi-electrode stimulation outcomes, although it still outperformed prior baseline models for stimulus threshold prediction. A key advantage of SBI, however, is that it provides a full posterior distribution over biophysical parameters for each cell, enabling principled uncertainty quantification. Because of model misspecification on real recordings, direct samples obtained by a forward pass through the neural posterior estimator had near-zero acceptance rate. We therefore used the learned posterior as a proposal distribution and drew posterior samples using MCMC.

To summarize posterior uncertainty across cells, we report the posterior contraction ratio, defined for each parameter as the ratio between the posterior standard deviation inferred by SBI and the standard deviation of the corresponding uniform prior. This quantity is averaged across the 37 cells for which SBI inference was performed. A contraction ratio close to 1 indicates that the data provide little information beyond the prior, whereas values substantially below 1 indicate strong constraint by

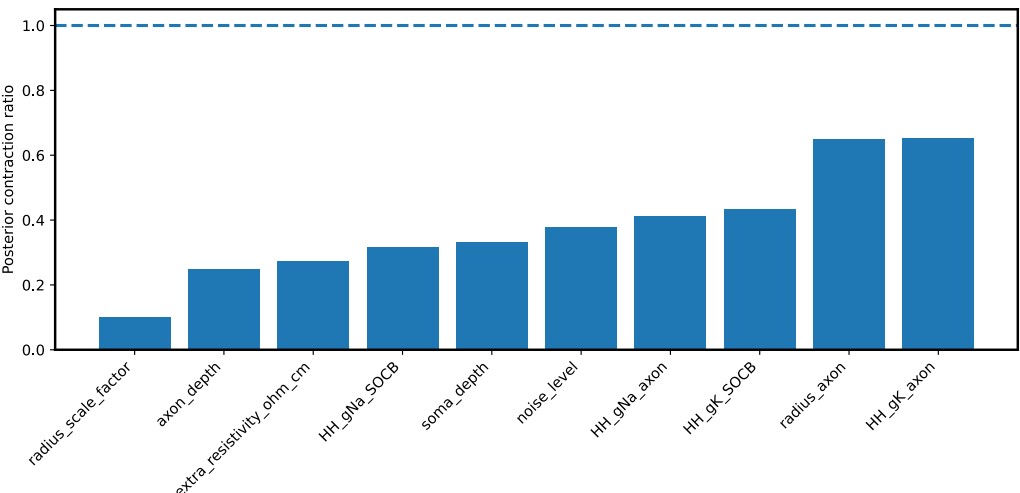

*Figure 17.* Posterior contraction ratios for all inferred biophysical parameters, averaged across 37 retinal ganglion cells. The contraction ratio is defined as the posterior standard deviation divided by the standard deviation of the corresponding uniform prior; values below one indicate posterior narrowing relative to the prior. Axonal potassium conductance and axon radius exhibit the weakest contraction, while geometric parameters such as the global radius scale factor and axon depth are more tightly constrained.

the observed EI features.

Figure 18 summarizes posterior contraction ratios across parameters. We observe that axonal potassium conductance and axon radius exhibit the weakest contraction, indicating higher uncertainty and partial non-identifiability from the available extracellular features. In contrast, parameters such as the global radius scale factor and axon depth relative to the electrode array are inferred with substantially greater precision. This pattern is consistent with the sensitivity of extracellular recordings: geometric parameters that directly modulate signal amplitude across many electrodes are more tightly constrained than conductances that primarily affect local spike dynamics.

Posterior uncertainty is not uniform across cells. Cells located closer to the MEA, or spanning a larger number of informative electrodes, exhibit substantially stronger posterior contraction across most parameters. This variability highlights an important advantage of SBI: uncertainty estimates adapt naturally to data quality and electrode coverage, providing cell-specific confidence measures that are unavailable from point-estimation methods alone.

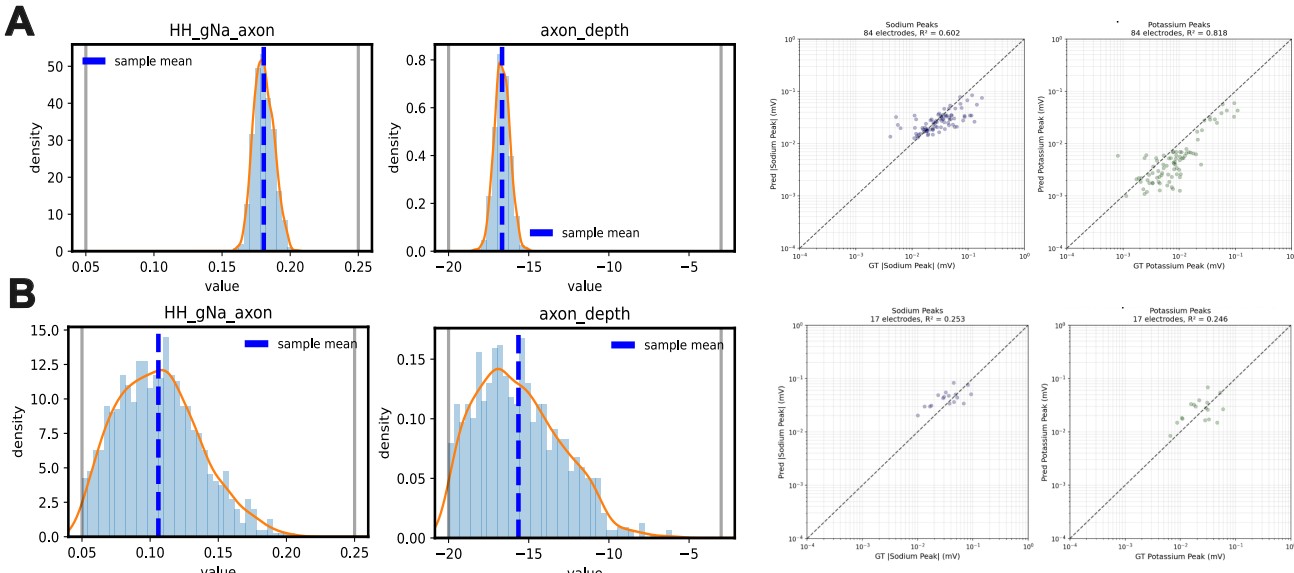

*Figure 18.* Posterior inference with SBI for two representative retinal ganglion cells (A and B). From left to right: posterior marginal distributions of axonal sodium conductance and axon depth, followed by comparisons of true and inferred sodium and potassium extracellular peak amplitudes. For extracellular peak comparisons, only electrodes within a local neighborhood of the inferred cell morphology are shown.

## S. On amortization for SBI on Real RGC Recordings

In the main text, SBI was applied in a non-amortized setting, where a separate neural posterior estimator was trained for each cell. A key advantage of simulation-based inference, however, is that the posterior estimator can be *amortized*: after a single training phase on simulations drawn from the prior, the same neural network can be reused to infer a posterior for any new observation without retraining. We exploit this property by representing both simulated and real cells in a shared *canonical pose*, enabling the posterior estimator to operate on a common observation space rather than on cell-specific coordinate systems.

Our simulator constructs each cell directly in this canonical pose: the soma is placed at the origin and the axon extends along the positive x-axis. We define the amortized prior directly in this canonical frame. It contains 10 varying biophysical parameters: the four SOCB/axon conductances, the axon radius, a global radius-scale factor, the extracellular resistivity, the soma and axon depths, and the noise scale $\sigma$ of the stochastic current injection. In addition, it includes $2N_{\text{cp}}^{\max}$ canonical axon control-point coordinates, where $N_{\text{cp}}^{\max} = 33$ is the maximum number of spline knots; shorter splines are zero-padded. The remaining biophysical parameters, to which the per-cell experiments were nearly insensitive, are fixed at the midpoints of their per-cell bounds.

For each simulated cell we compute its EI on a fixed 512-channel canonical template, and this EI provides the per-electrode features seen by the network. The observation vector includes only EI-derived features and *excludes stimulation thresholds*, which are used in the per-cell inference pipeline: thresholds are measured at cell-specific, experimentally chosen stimulation sites and would therefore need to be predicted in the canonical frame for each candidate parameter vector $\theta$, requiring an expensive binary search inside each simulation and substantially increasing computational cost. Each canonical electrode additionally carries a binary mask channel that is itself fed to the network as an input feature alongside the EI features, so the network also receives, per electrode, a flag indicating whether that value is informative. The mask retains only electrodes close enough to the cell to carry meaningful signal: a simulated electrode counts as near the cell if its EI peak exceeds a fixed threshold of $10\,\mu\text{V}$, while far electrodes receive only tiny passive signals and are masked.

At test time we map each real cell into this same canonical pose. From the cell's EI we identify which electrodes lie on the axon; the axon direction is then defined as the principal axis of the 2-D positions of these axon electrodes together with the soma electrode, oriented to point away from the soma. The cell is translated so that its soma lies at the origin and rotated so that its axon aligns with the positive x-axis, matching the simulation convention. Its 512 measured electrodes, expressed in this canonical frame, are mapped onto the canonical template by nearest-neighbor assignment. As in simulation, each real

cell is also described by a mask channel, and the mask is again supplied to the network as an input feature. The real-cell mask serves the same purpose as in simulation, but must additionally account for incomplete array coverage: a template position is retained only if both (i) some measured electrode lies within $20\,\mu$m of it and (ii) the measured EI peak there exceeds the same $10\,\mu$V threshold; failure of either condition masks the entry. The two criteria are complementary: the coverage check asks whether the array recorded anything at that position, while the signal threshold asks whether the cell is close enough for the recording to carry useful signal, applying the same definition of proximity used in simulation. We acknowledge that a failure case of our approach arises for cells near the MEA boundary, where the coverage criterion masks positions close to the cell and yields a mask pattern unlike any seen in simulation, degrading the posterior fit for these cells.

We draw $10^5$ simulations from this canonical prior and train a single normalizing-flow neural posterior estimator on them. At test time, we condition this estimator on each cell's canonical-frame features and draw $10^3$ posterior samples. Posterior summaries are then transformed back into the original cell-specific coordinate frame by applying the inverse rotation and translation, thus allowing a single trained network to yield posterior estimates for new cells.

Qualitatively, the amortized SBI results closely match the per-cell SBI results shown in the main text. The scatter plots in Fig. 19 have a similar visual structure to those obtained with non-amortized SBI (Fig. 16): the point clouds show comparable alignment with the diagonal, and the same features exhibit the largest departures from perfect prediction. We note, however, a slight downward bias for the amortized version for the three different peaks.

We next quantify this agreement in Table 11. We report both the plain Pearson $R^2$ and an IQR residual down-weighted $R^2$. This distinction is important because $R^2$ is highly sensitive to outliers: a small number of large residuals can make the score poor even when the scatter plot shows a clear overall trend. To better reflect the typical predictive performance of the method, we therefore also report the down-weighted metric, which reduces the influence of these extreme residuals. Under this metric, amortized SBI achieves positive $R^2$ for the sodium and potassium EI amplitudes, with values of $0.312$ and $0.426$, respectively, and smaller but positive values for propagation velocity and spike duration. The sodium and potassium scores are higher than in the per-cell SBI table, but this primarily reflects the choice of summary metric: the amortized scatter plots show a clearer linear trend with the observed amplitudes, even though the predictions remain somewhat biased.

Finally, the amortized posterior remains effective for downstream multi-electrode spike-activation prediction. As shown in Table 12, amortized SBI achieves an accuracy of $0.894$ and a balanced accuracy of $0.871$ for triplet threshold classification. This performance is close to the per-cell Hodgkin–Huxley fits and compares favorably with the per-cell SBI model and the non-biophysical baselines.

In conclusion, it appears that amortization does not hurt the performance of SBI.

*Table 11.* Feature prediction performance of amortized SBI on real RGCs, reported as $R^2$ between observed and model-predicted features. We report both plain Pearson $R^2$ using all finite points and IQR residual down-weighted $R^2$.

| Metric | Sodium | Potassium | Capacitive | Velocity | Duration | Stim. Thresholds |
|---|---|---|---|---|---|---|
| Plain $R^2$ | 0.004 | 0.157 | -0.263 | -0.922 | -0.457 | -0.297 |
| IQR down-weighted $R^2$ | 0.312 | 0.426 | -0.132 | 0.036 | 0.093 | -0.130 |

*Table 12.* Spike-activation classification performance of amortized SBI on real RGCs. Accuracy and balanced accuracy are reported together with false-positive and false-negative rates.

| Method | Acc. ↑ | Bal. Acc. ↑ | FPR | FNR |
|---|---|---|---|---|
| Amortized SBI | 0.894 | 0.871 | 0.066 | 0.192 |

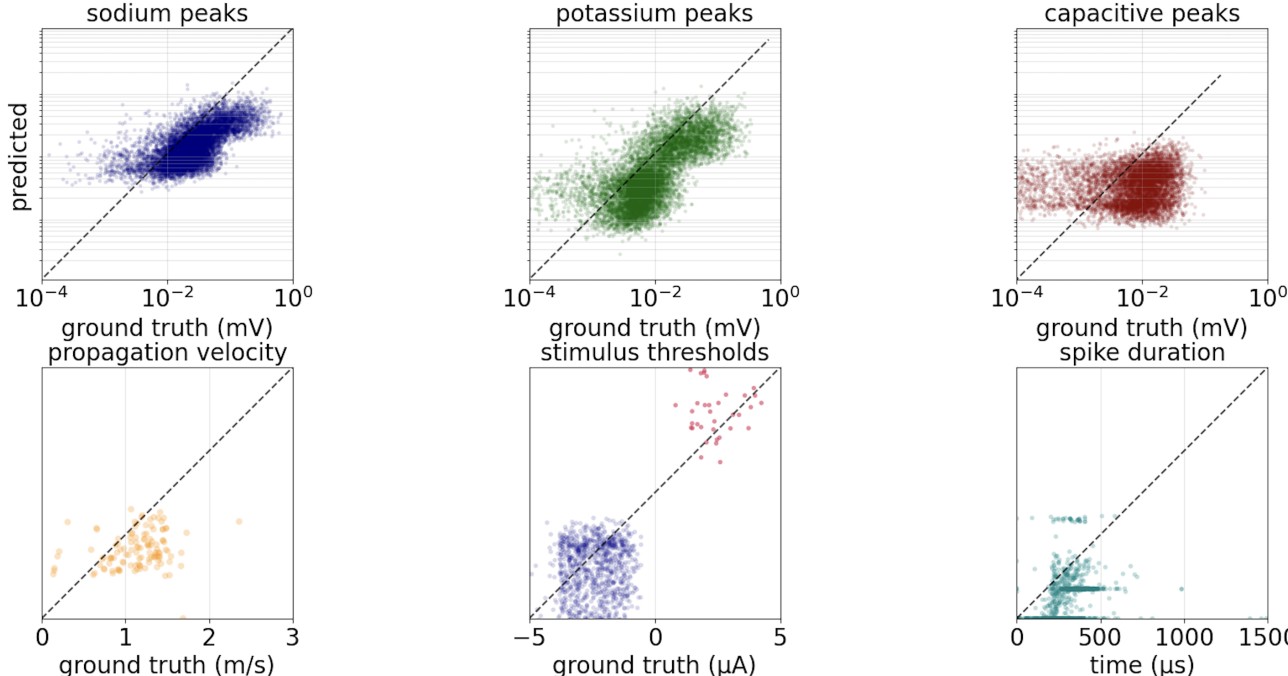

*Figure 19.* Extending figure 4, models trained with the amortized version of SBI on real RGCs were then used to predict the EI features. Amortized-SBI trained models matched per-cell SBI in trends in its capturing of EI features.

*Figure 20.* Three-electrode stimulation response map for an example RGC (same cell as the top row of Figure 5). Yellow markers indicate stimulus combinations that evoked spiking; HH (GD) predictions are compared to superposition and independent-site baselines.

*Table 13.* Multi-electrode prediction accuracy on 37 cells. HH and heuristic rows pool all stimuli. MLP uses held-out *cells* (7/37 test, 20 random splits; mean test metrics).

| Method | Acc. ↑ | Bal. Acc. ↑ | FPR | FNR |
|---|---|---|---|---|
| HH (GD, sodium + stimulation) | **0.904** | **0.893** | 0.077 | 0.137 |
| HH (GD, balanced features) | 0.892 | 0.886 | 0.098 | 0.131 |
| HH (SBI, per-cell) | 0.873 | 0.841 | 0.072 | 0.245 |
| MLP (held-out cells) | 0.859 | 0.833 | 0.107 | 0.228 |
| Superposition baseline | 0.843 | 0.811 | 0.102 | 0.276 |
| Independent-site baseline | 0.854 | 0.812 | 0.074 | 0.302 |

## T. Multi-Electrode Stimulus Analysis and Model Limitations

This appendix supplements the main-text multi-electrode results (Table 1, Figure 5). We work exclusively with the 37 parasol cells that have full *three-electrode* stimulation maps (Appendix O). Throughout, experimental responses and model predictions are treated as binary spike/no-spike outcomes by thresholding empirical spike probability at 0.5, and population metrics aggregate performance over all stimulus combinations in all cells.

### T.1. Three-electrode response maps

Figure 5 shows two-dimensional cross-sections of the three-electrode stimulus space for clarity. The experimental protocol delivered simultaneous currents on all three selected electrodes; Figure 20 shows the full three-electrode map for the top-row example cell from the main text.

### T.2. Three HH methods and reference baselines

We compare three HH fitting pipelines, all trained *without* multi-electrode labels:

- **GD (sodium + stimulation):** gradient descent emphasizing sodium peak amplitudes and stimulation thresholds (triplet-trained checkpoint; main-text Table 1).

- **GD (balanced features):** gradient descent with balanced EI feature weights (Figure 4 on 198 cells).

- **SBI:** per-cell simulation-based inference (Appendix R).

We do not report amortized SBI here (Appendix S).

Table 13 also reports reference predictors on the same 37 cells and stimulus grid: the superposition and independent-site heuristics, and a data-driven MLP fit directly to multi-electrode labels (next subsection). Surprisingly, GD (sodium + stimulation) attains the highest pooled accuracy (0.904) despite Figure 4 using the balanced-features checkpoint—consistent with multi-electrode prediction being dominated by axon *location* and local excitability rather than by matching every EI feature with equal weight.

### T.3. Data-driven MLP baseline

A natural question is whether gains over simple heuristics require a biophysical forward model or only greater model capacity. We therefore trained a supervised multilayer perceptron on multi-electrode *labels* with the same held-out cell-split protocol used in the main text. The network sees, for each stimulus combination, normalized currents at the three stimulating electrodes plus EI-derived features at those sites (15 dimensions after featurization); labels are binary spike/no-spike at probability 0.5. Crucially, training uses multi-electrode responses from 27 cells while evaluation uses 7 held-out cells the model never sees, repeated over 20 random cell-level splits (seeds 42–61).

The MLP ($128 \rightarrow 64 \rightarrow 1$, ReLU, dropout) fits the *training* cells very well: mean accuracy $0.936 \pm 0.035$ and balanced accuracy $0.921 \pm 0.045$ across splits. On held-out test cells it drops to $0.859 \pm 0.033$ accuracy and $0.833 \pm 0.048$ balanced accuracy—below HH (GD, sodium + stimulation) on every metric despite having direct access to thousands of multi-electrode labels from training cells, an information advantage HH never receives during fitting.

This train–test gap indicates that *biophysical inductive bias*, not model capacity alone, drives generalization to new cells from single-electrode data. The MLP can memorize multi-electrode response geometry within the training distribution but does not transfer the extrapolation HH achieves by enforcing membrane dynamics and spatially extended axon structure. We emphasize that this comparison is intentionally favorable to the data-driven model: HH is evaluated on all 37 cells with no multi-electrode supervision, whereas the MLP is scored only on held-out cells *after* seeing multi-electrode maps from most of the cohort.

### T.4. Per-cell analysis: GD (sodium + stimulation)

Pooled metrics hide cell-to-cell variability (GD per-cell accuracy mean 0.907, SD 0.06). We therefore ask how *stimulation site* affects prediction accuracy. Using the recorded EI, we assign each cell a label indicating whether its three-electrode stimulation triplet lies at the sodium channel band (SOCB) of the cell or further along the axon. This yields 14 SOCB-labeled cells (triplet at the SOCB) and 23 axon-labeled cells (triplet displaced along the axon).

Figure 21 plots per-cell GD accuracy against each heuristic baseline, color-coded by stimulation site. For SOCB-labeled cells: GD mean 0.932, superposition 0.888, independent-site 0.854. For axon-labeled cells: GD mean 0.892, superposition 0.825, independent-site 0.860. When the triplet sits at the SOCB, normalized currents sum at one highly excitable site and the superposition model is nearly as accurate as HH (left panel, points near the diagonal). When the triplet lies further along the axon, both heuristics degrade and the biophysical model separates further (right panel).

### T.5. Limitations revealed by feature recovery and multi-electrode performance

Figure 4 shows that our fitted models recover extracellular features only partially—moderate $R^2$ on several supervised quantities despite strong multi-electrode prediction—which clarifies where the forward model still falls short of a fully faithful biophysical *digital twin*. Our long-term goal remains a model that reproduces both EI features and stimulation responses from the same parameters; the discussion below separates limitations exposed by feature mismatch from the complementary fact that multi-electrode accuracy can remain high when prediction is dominated by geometry and local excitability at stimulated sites.

**Measurement limitations.** One obstacle to a biophysical forward model that reproduces both extracellular features and stimulation responses with very high fidelity is that our stimulation hardware is not perfectly reliable. Electrode-to-electrode shunting and calibration variability mean that the current delivered at one site can differ from the current assumed by the forward model, especially in $\sim 1000~\Omega{\cdot}$cm saline, whose resistivity is comparable to that of the retina. Across cells, the per-electrode scale factors required to align predicted and measured single-electrode thresholds typically span roughly 0.5–0.85. We mitigate this mismatch by estimating, for each cell and stimulating electrode, a current scale factor that maps commanded stimulus amplitude to effective amplitude in the model, chosen so that the model's single-electrode threshold matches the measured threshold; multi-electrode predictions then use these scaled currents, using only the same single-electrode threshold data already employed during fitting. Separately, mapping heterogeneous retinal resistivity remains limited by MEA pitch: imaging resolution scales with electrode spacing, so a 30 $\mu$m array cannot resolve fine inhomogeneity—an active measurement direction in our lab.

**Model expressiveness.** The biophysical template must eventually incorporate more parameters and fewer rigid regional assumptions to approach a *digital twin* capable of arbitrary stimulation waveforms beyond the triphasic triplets tested here.

Multielectrode stimulation prediction per-cell accuracy

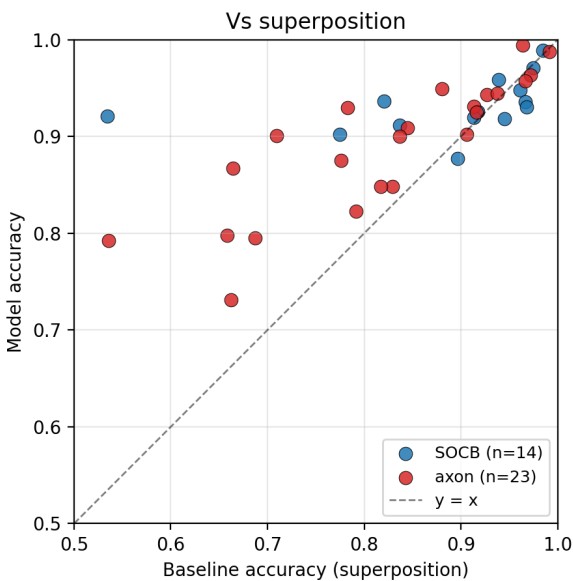 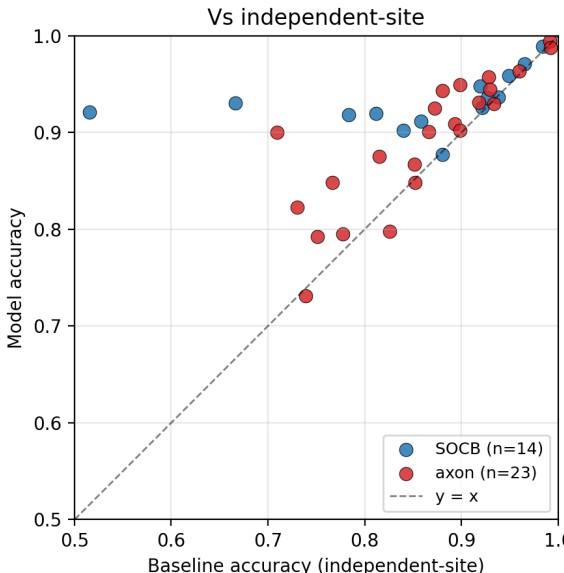

*Figure 21.* Multielectrode stimulation prediction per-cell accuracy for HH (GD, sodium + stimulation) versus superposition (**left**) and independent-site (**right**) baselines. Each point is one of 37 cells. Blue: stimulation triplet at the SOCB; orange: triplet further along the axon. Dashed line: equal accuracy.

