# OpenReview forum: "Learning Biophysical Models of Large-Scale Multineuronal Data To Enable Precise Neurostimulation"
_ICML.cc/2026/Conference — ICML 2026 spotlight_

### Official Review · Reviewer_UXxY · 2026-03-09

**Soundness:** 3
**Presentation:** 3
**Significance:** 4
**Originality:** 3
**Overall Recommendation:** 5
**Confidence:** 4

**Summary:**

This paper introduces a framework for inferring cell-specific multi-compartment Hodgkin-Huxley (HH) biophysical parameters from high-density extracellular multi-electrode array (MEA) recordings, without requiring invasive intracellular measurements. The core technical contributions are: (1) a set of differentiable, physically interpretable features extracted from the electrical image (EI) of each neuron, capturing sodium/potassium/capacitive peak amplitudes, spike duration, and inter-electrode propagation delays, that enable stable gradient-based optimization via automatic differentiation in the JAXLEY simulator; (2) a novel differentiable relaxation of binary stimulation thresholds, allowing stimulation-based constraints to be used in gradient descent; and (3) a simulation-based inference (SBI) pathway that provides a full approximate posterior over biophysical parameters under a stochastic forward model. The method is validated on simulated and real data, including 207 parasol RGCs from 14 isolated macaque retina preparations. Models fit from just a few minutes of single-electrode recording data achieve 90.4% accuracy in predicting spike/no-spike outcomes under unseen simultaneous multi-electrode stimulation patterns, outperforming prior baselines.

**Compliance With Llm Reviewing Policy:**

Affirmed.

**Ethical Review Concerns:**

This paper uses recordings from isolated macaque retinas, involving non-human primate tissue. The paper does not mention IRB approval, institutional animal care oversight, or tissue sourcing details. Given that the clinical goal is to inform retinal prosthetic devices for human patients, an ethics reviewer should confirm that appropriate oversight was in place. The authors should also briefly address the pathway to human clinical use and any associated safety considerations.

**Ethical Review Flag:**

Flag this paper for an ethics review.

**Ethics Expertise Needed:**

["Inappropriate Potential Applications & Impact (e.g., human rights concerns)", "Responsible Research Practice (e.g., IRB, documentation, research ethics)"]

**Final Justification:**

After reading the author rebuttal and the responses to all reviewers, I maintain my recommendation of 5: Accept and raise my confidence to 4.

**Key Questions For Authors:**

1. The SBI pathway yields a full approximate posterior, but the only validation reported is the posterior contraction ratio. I could not find a simulation-based calibration or coverage test in the paper. Could the authors clarify whether such a check was performed, and if not, provide one? This would help assess whether the approximate posterior is reliable when applied to real tissue data that may differ from the training prior. Additionally, the posterior mode is not generally the optimal Bayes estimator for a downstream classification task. Would using a posterior-averaged prediction (e.g., integrating $f_\mathrm{stim}$ over posterior samples) improve SBI accuracy beyond the reported 88.8%? Could the authors also clarify what drives the underperformance of SBI relative to gradient descent?

2. Both the superposition and independent-site baselines are structurally constrained models. I was not able to assess from the paper alone how much of the gain comes from the biophysical inductive bias versus having a more expressive model. Would a data-driven approach (e.g., logistic regression or a small neural network trained on single-electrode threshold data) significantly underperform the HH model?

3. The duration-related EI feature is reported to be substantially more misspecified in real tissue, but its $R^2$ is not given in the main text. What is this value, and could this misspecification explain a meaningful share of the approximately 10% prediction errors in the multi-electrode task? I found it difficult to draw a clear connection between feature-level fit quality and downstream prediction errors from the information provided.

**Limitations:**

The authors discuss limitations honestly, covering forward model fidelity, timing-related EI feature misspecification, and the absence of synaptic coupling. The discussion of limitations is appropriate and reflects well on the honesty of the work.

**Strengths And Weaknesses:**

**Soundness**

The paper is technically sound. The HH forward models are clearly stated, and the line-source approximation for extracellular potentials is well-established and appropriate. The differentiable EI feature extraction and the sigmoidal relaxation of binary stimulation thresholds are well-motivated engineering choices that enable gradient flow through an otherwise non-smooth problem. The simulated experiments are well-designed, and the feature ablation study (Appendix I) usefully identifies which observables constrain the parameters, which is a key question from a system identification perspective.

There are several points that I found difficult to fully assess. First, the paper offers two inference modes: gradient descent yields a point estimate of $\theta$, while SBI yields a full approximate posterior from which the posterior mode is extracted for downstream prediction. The only posterior validation reported is the posterior contraction ratio (Appendix P), which indicates whether the data are informative relative to the prior, but I could not find a simulation-based calibration check or coverage test, which would confirm whether true parameters fall within credible intervals at the right frequency. It is not clear to me whether this omission reflects a deliberate choice or a limitation, but it leaves the reliability of the approximate posterior uncertain, particularly when applied to real tissue data that may differ from the training prior. Second, $R^2$ values for feature-fitting on real data are only moderate ($R^2 = 0.417$ for capacitive peak; $R^2 = 0.383$ for propagation velocity), and I could not understand from the paper why these moderate fit qualities still lead to 90.4% downstream accuracy. A more explicit discussion of this would help readers assess the robustness of the approach. Third, I found it difficult to evaluate the added value of the biophysical inductive bias without a data-driven baseline (e.g., a small neural network trained on single-electrode thresholds) for comparison.

**Presentation**

The paper is clearly written and well structured. One weakness is that SBI results are largely relegated to appendices, while SBI is presented as a key contribution. More prominent reporting in the main text would strengthen the paper. I also could not find a clear explanation of why SBI underperforms gradient descent (88.8% vs 90.4%) on the primary prediction task, and this left me uncertain about how to interpret the role of SBI in the overall framework.

**Significance**

This work addresses a practically important problem: replacing hours of empirical stimulation testing with model-based predictions from minutes of passive recordings. This has direct relevance to retinal prosthetics and, more broadly, to closed-loop neural implants. A calibrated forward model of this kind is a prerequisite for any principled stimulation design, and the paper makes a meaningful step in that direction. The methodological contributions are likely to be useful beyond the retinal application.

**Originality**

The combination of differentiable simulation, engineered EI features, and SBI for population-level HH fitting from real extracellular data is novel. Individual components exist in prior work, but their integration and application to large-scale real recordings for neurostimulation prediction has not been demonstrated before. The differentiable threshold relaxation is a small but genuine technical contribution. One caveat is that the RGC model architecture is directly adapted from prior work (Vilkhu et al., 2025; Kish et al., 2023), and the paper relies heavily on existing infrastructure (JAXLEY, Kilosort2). The novelty is in the inference methodology, and this could be stated more clearly.

---

> ### Author Rebuttal · Authors · 2026-03-31
>
> We thank Reviewer UXxY for the thorough review and kind words in the limitations discussion. We address each point below.
>
> **[Ethics — Animal use and clinical pathway]** We will include a wet lab methods section in the final version with full IRB and tissue sourcing details. Briefly, macaque retinal tissue was obtained from terminally anesthetized animals euthanized during research performed by other laboratories, in accordance with IACUC approval and national guidelines. The immediate translational goal is reducing empirical calibration burden for epiretinal prostheses; safety and regulatory considerations for clinical deployment will be addressed in future work.
>
> **[Q1 — SBI calibration and posterior averaging]** A coverage check was performed for the straight-axon simulated setting (Appendix J, Figure 10B), where ground-truth parameters are available, showing that true parameters consistently fall within $\pm 2$ posterior standard deviations across noise levels. However, the reviewer is correct that no such calibration check was performed for real RGC data, where ground truth is unavailable by definition. The posterior contraction ratios in Appendix P therefore serve as a proxy, indicating data informativeness relative to the prior rather than confirming calibration. We will make this distinction explicit in the final version. Regarding posterior averaging: the posterior mode was used for computational convenience, but integrating predictions over posterior samples could in principle improve accuracy by accounting for parameter uncertainty rather than committing to a point estimate. We have not evaluated this in the current work but agree it is a promising direction. We address the underperformance of SBI relative to GD in our response to reviewer h6Bp.
>
> **[Q2 — Superior data-driven baselines]** In response to concerns that our baselines were too narrow, we implemented two additional data-driven baselines: (1) logistic regression and (2) a small MLP (128 to 64 to 1 with dropout), both taking as input the same EI-derived features used to fit the HH model: sodium, diffusion, and potassium peak amplitudes and pairwise sodium peak timing differences at the three stimulating electrodes, plus current amplitudes normalized by the single-electrode thresholds, which encodes the single-electrode stimulus threshold information directly. Both models were trained on multi-electrode stimulation responses from 27 cells, validated on 3 cells, and evaluated on 7 held-out cells unseen during training. The MLP achieves strong in-distribution performance (train acc=0.919, bal. acc=0.912), confirming it can fit multi-electrode response patterns when training data is available. However, it generalizes poorly to held-out cells (test acc=0.867, bal. acc=0.789), underperforming HH (GD) (acc=0.904, bal. acc=0.893) and falling only modestly above the superposition baseline (bal. acc=0.811). The logistic regression degenerates further (test acc=0.803, bal. acc=0.489), below even the independent-site baseline (bal. acc=0.812), essentially predicting the majority class on low spike-rate test cells, a failure mode the biophysical model avoids entirely. The inability of the MLP to generalize beyond the training distribution reinforces our claim about the necessity of the biophysical inductive bias. We note this comparison is not perfectly apples-to-apples: our HH model observes zero multi-electrode responses during fitting, while these baselines were trained on multi-electrode labels from 27 cells, a substantial information advantage HH never has. We also note that the superposition model is already closely related to a constrained GLM: it predicts spiking when the sum of normalized currents exceeds one, which is a linear threshold in the same normalized current space used by our logistic regression baseline, with weights fixed to one rather than learned. Even with the additional advantage of learned weights and multi-electrode supervision, the data-driven models underperform HH on every metric, demonstrating that the performance gain of our approach stems from the biophysical inductive bias rather than model expressiveness.
>
> **[Q3 — Duration mismatch and connection to prediction errors]** All $R^2$ values for GD and SBI are in Appendix Table 10; the duration feature is labeled "width" (we will fix this). Spike duration $R^2=0.229$. We will add a section in the final version making explicit the intuitive connections between fitted features and downstream prediction accuracy. Briefly, multi-electrode prediction is primarily determined by axonal orientation and location relative to the electrodes, illustrated by the two extreme cases in Figure 5. Spike duration encodes ion channel kinetics and membrane capacitance rather than axon geometry, so its misspecification likely has little bearing on prediction accuracy. Residual errors are more likely attributable to axon orientation misestimation from inhomogeneous tissue resistivity.

---

> > ### Author Rebuttal · Reviewer_UXxY · 2026-04-03
> >
> > I thank the authors for their detailed and thorough rebuttal. My three main concerns have been addressed as follows. I maintain the score I indicated in my original review, which I believe is fair given the quality of the work and the satisfactory rebuttal.

---

### Official Review · Reviewer_mLRk · 2026-03-11

**Soundness:** 4
**Presentation:** 3
**Significance:** 4
**Originality:** 3
**Overall Recommendation:** 6
**Confidence:** 3

**Summary:**

The paper proposes a method to fit multi-compartment Hodgkin-Huxley model parameters based on multi-electrode array (MEA) recordings. This would allow to create a digital twin of the neural network on MEA system, which could be used to predict network response to specific patterns of neurostimulation. The proposed approached is first thoroughly described and then tested on both simulated and experimentally acquired data. Simulated data have a ground truth, which allows for a quantitative analysis of the performance of the proposed approach. Experimental data, although they do not come with a ground truth, allow to assess the performance of the approach in a real environment. Experimental results show very promising performance.

**Compliance With Llm Reviewing Policy:**

Affirmed.

**Final Justification:**

This is an excellent paper that tackles a very important and timely problem in neuroscience with state-of-the-art methodology.

**Key Questions For Authors:**

I do not have specific question for the authors.

**Limitations:**

Yes

**Strengths And Weaknesses:**

This is an excellent paper that tackles a very important and timely problem in neuroscience with state-of-the-art methodology.

---

> ### Author Rebuttal · Authors · 2026-03-31
>
> We thank the reviewer for taking the time to review our submission and for their encouraging assessment of our work. We are glad the contribution and experimental validation were compelling.

---

> > ### Author Rebuttal · Reviewer_mLRk · 2026-04-02
> >
> > I find the authors' responses to the different reviewers' comments appropriate.

---

### Official Review · Reviewer_DdHr · 2026-03-12

**Soundness:** 3
**Presentation:** 2
**Significance:** 3
**Originality:** 3
**Overall Recommendation:** 4
**Confidence:** 4

**Summary:**

This paper presents a framework for fitting multi-compartment Hodgkin–Huxley models of individual neurons directly from extracellular high-density MEA data. The key idea is to avoid direct waveform matching and instead fit differentiable, biophysically motivated EI features together with a differentiable surrogate for single-electrode stimulation thresholds, allowing gradient-based optimization in JAXLEY and complementary simulation-based Bayesian inference for uncertainty quantification. The method is evaluated on progressively more realistic settings, from straight-axon simulations to realistic retinal ganglion cell morphologies and finally to macaque retina recordings. On real parasol cells, the fitted models are then used to predict previously unseen multi-electrode stimulation responses, outperforming superposition and independent-site baselines. Overall, this is an ambitious and thoughtful paper that tackles an important bottleneck in scalable biophysical modeling from extracellular data.

**Compliance With Llm Reviewing Policy:**

Affirmed.

**Key Questions For Authors:**

1. It would be helpful to report more clearly how much single-electrode stimulation data are required per cell in practice. How many electrodes and polarities were typically measured, how many repeats were used to estimate the empirical 50% threshold, and what is the approximate experimental time budget per cell?

2. Please report computational cost in a more concrete way. What are the wall-clock runtimes and hardware requirements per cell for gradient-based fitting and for SBI, how many simulator calls are required, and how many restarts were typically needed for stable convergence?

3. The real-data evaluation would be much more informative with per-cell distributions and failure analysis, rather than only aggregate accuracy over all tested outcomes. Are the remaining errors concentrated for specific cell geometries, near particular axonal regions such as the hillock or sodium channel band, or for particular multi-electrode configurations?

4. More broadly, given that the paper already identifies model misspecification as an important limitation, it would be interesting to hear how the authors think about extending this framework toward inference methods that are more robust to simulator mismatch and perhaps more directly aligned with downstream stimulation design.

**Limitations:**

Yes.

**Strengths And Weaknesses:**

**Strengths**

1. The paper addresses a genuinely important problem. Fitting detailed HH-type models usually requires intracellular measurements, which are low-throughput and impractical at population scale. Here the authors show a plausible route toward recovering interpretable geometry and conductance parameters from extracellular measurements that are much easier to acquire at scale. That is a significant conceptual step.

2. A major strength is the design of the objective. Direct waveform-level comparison appears to be poorly behaved even in the simple straight-axon setting, whereas the proposed feature-based loss produces substantially better recovery. The use of differentiable EI features and a smooth surrogate for stimulation thresholds is simple, well motivated, and central to making gradient-based fitting feasible. The ablation analysis also helps support that this is not just an implementation detail.

3. The experimental progression is also strong. The paper builds evidence from controlled simulations, realistic RGC morphologies, and then real retinal recordings. The final downstream test on unseen multi-electrode stimulation is especially compelling because it evaluates genuine out-of-training generalization rather than only fit quality on the same summary features used for inference.


**Weaknesses**

1. The baseline comparison is somewhat narrow for the final prediction task. The paper compares mainly against two mechanistically motivated baselines, superposition and independent-site models, which represent limiting assumptions about electrode interactions. These are relevant baselines, but they are also fairly simple. The empirical case would be stronger if the authors also compared against learned but supervision-matched predictors that use the same single-electrode threshold information and possibly EI-derived geometry features, for example a constrained GLM or another low-capacity compositional predictor. As written, the results clearly establish improvement over simple heuristic models, but they do not yet fully establish superiority over stronger predictive alternatives under the same information budget.

2. A second concern is that the inference problem is made tractable partly by fixing many biophysical quantities across cells and preparations. The inferred parameter vector contains conductances, radii, and geometry, but membrane capacitance, reversal potentials, ion-channel kinetics, initial conditions, and some additional channel types are held fixed from prior work. This is understandable, but it means the quality and interpretation of the recovered parameters depend on a strong set of structural assumptions. In real tissue, some of the apparent parameter recovery may reflect compensation for misspecified fixed quantities. A more systematic sensitivity analysis would help clarify which conclusions are robust and which depend on these choices.

3. The extracellular forward model is also simplified in ways that may limit fidelity, particularly for timing features. The EI model uses a line-source approximation with homogeneous conductivity, while extracellular stimulation is modeled via point sources in a homogeneous conductive half-space. These approximations are standard and probably sufficient for coarse behavior, but the paper’s own real-data analysis shows that amplitude-based features are fit more reliably than timing- and width-related features. This suggests that at least part of the remaining error may come from forward-model mismatch rather than only optimization difficulty. That issue deserves more discussion because the timing features are also among the most biophysically informative constraints.

4. There are also some presentation issues. In particular, the units used for EI amplitudes appear inconsistent across figures: the real EI plots are shown in microvolts, while the simulated waveforms in Figure 3 are labeled in millivolts. This may be a plotting choice, but it should be clarified. I also noticed a few minor notational rough edges, though none of them affect the main conclusions.

---

> ### Author Rebuttal · Authors · 2026-03-31
>
> We thank Reviewer DdHr for the careful and constructive review. We address each point below.
>
> **[W1 — Baseline Comparisons]** Reviewer UXxY raised identical feedback; we address it in detail in our response to their review (Q2). Briefly, we trained logistic regression and MLP baselines on multi-electrode responses from 27 cells and evaluated on 7 held-out cells, an information advantage HH never has. Both underperform HH (GD) (bal. acc=0.893): MLP achieves 0.789 and logistic regression 0.489.
>
> **[W2 and W3 — Fixed parameters and forward model approximations]** Our synthetic data experiments demonstrate that the inference framework correctly recovers parameters when the forward model is well-specified, including a larger class of parameters than prior extracellular inference work [Tanoh et al., 2025; Gold et al., 2007]. The feature mismatch observed in Figure 4 on real data, which we acknowledge in the discussion, is therefore most naturally explained by forward model mismatch rather than inference failure. We do not claim identifiability in the real data setting. In this work we focused on axon orientation, location, ion channel densities, and radii, as these are most variable across preparations and most impactful on the stimulus prediction task [Kish et al., 2023; Vilkhu et al., 2025]. The most significant limitation in the fidelity of the forward model currently is that we model the extracellular medium as homogeneous, following convention in retinal modeling literature. We are pursuing parallel work to directly measure heterogeneous retinal resistivity using resistance tomography, and intend to expand the discussion of forward model limitations substantially in the final version.
>
> **[W4 — Units inconsistency]** Thank you for flagging this. We can switch all EI amplitude axes to $\mu$V in the final version for consistency.
>
> **[Q1 — Single-electrode data per cell]** The time budget is defined per electrode rather than per cell. For each of the 512 electrodes, 25 repeated trials were delivered at each of 40 current amplitudes spanning 0 to 4 $\mu$A, with a minimum inter-trial interval of ~10 ms, yielding ~90 min total for the full preparation. The majority of stimulation data was collected using cathodic pulses (a more detailed data collection description will be added as requested by Reviewer h6Bp). We note that most stimuli well above or below threshold contribute little information about the threshold itself, and parallel work in our lab suggests that all single-electrode thresholds could be extracted in ~3 minutes using targeted closed-loop stimulation.
>
>
>
> **[Q2 — Computational cost]** Gradient-based fitting was run on a CPU-only machine (Intel Xeon Gold 6240R @ 2.40 GHz; JAX on CPU). After compilation, the steady-state runtime was 1.16 s/epoch for a representative real cell in triplet mode, with stable convergence requiring up to 1000 epochs per cell (19 minutes total). Peak resident memory during training was 4.6 GB. No restarts were required. A single initialization from the EI-based location estimate was sufficient for stable convergence in all evaluated cells. SBI training required 10,000 simulator calls per cell, with a steady-state cost of 0.0285 s per EI simulation and 0.073 s per stimulus threshold evaluation per electrode, and an average wall-clock NPE training time of 225.1 s per cell (4 minutes total). Full hardware specifications will be reported in the final version.
>
> **[Q3 — Per-cell failure analysis]** We performed a per-cell failure analysis of multi-electrode stimulus prediction accuracy on all 37 real cells and will include per-cell distributions and grouped breakdowns in the revised paper. A notable pattern is that prediction accuracy is higher when at least one stimulating electrode falls near the fitted SOCB location: mean accuracy is
> 0.9375±0.0325 (mean±sd) for the 21/37 near-SOCB cells versus
> 0.8667±0.0652 for the remaining 16/37 cells. More than half of cells have at least one stimulating electrode near the SOCB, consistent with it being the most excitable region. We hypothesize this arises because SOCB stimulation is relatively insensitive to localization errors due to the region's spatial compactness and high sodium channel density, whereas axonal stimulation depends critically on precise 3D axon orientation, so small trajectory estimation errors translate directly into prediction errors.
>
>
> **[Q4 — Robustness to simulator mismatch]** First, robustness to simulator mismatch could be improved by incorporating misspecification-aware SBI variants that account for distributional shift between simulated and real observations, e.g. by adding noise to the simulated features [Bernaerts et al., 2025]. Second, aligning inference more directly with downstream stimulation design could be achieved by extending the framework to incorporate stimulation responses as additional supervision during fitting, rather than relying solely on single-electrode thresholds.

---

> > ### Author Rebuttal · Reviewer_DdHr · 2026-04-03
> >
> > I thank the authors for their rebuttal. They have answered or explained almost all of my questions/concerns, and I would like to ask them to make sure the promised additions to the discussion and some more clarification on methods, etc., in response to my questions are added to the final manuscript.

---

> > > ### Author Response · Authors · 2026-04-08
> > >
> > > We thank Reviewer DdHr for the thorough and constructive review. The additional baselines in particular make our paper much stronger. We will ensure all promised additions make it into the final draft.

---

### Official Review · Reviewer_h6Bp · 2026-03-13

**Soundness:** 4
**Presentation:** 4
**Significance:** 4
**Originality:** 4
**Overall Recommendation:** 5
**Confidence:** 5

**Summary:**

The paper describes a method to fix a small population of HH neurons to electrode array recordigs in vitro. The neurons are simulated in jax using multiple compartments of HH models. The models are fitted to the electrode recordings via gradient descent and SBI.

In simulated data with simulated neurons and MAE, the fit enable to recover neuron parameters. The fit enables to recover the soma radius and some conductance for instance. The fit is remarkably good with SBI. With in-vitro data the model is capable to infer stimulating electrode patterns.

**Compliance With Llm Reviewing Policy:**

Affirmed.

**Key Questions For Authors:**

In table 1, it seems that GD is better for this problem. Which is in contradiction with the Figure 3 where SBI is better.
Is there something to understand or comment on the pro and cons of the two methods? Do we understand why those results are different?

**Limitations:**

See weaknesses.

**Strengths And Weaknesses:**

Strengths:

1) Fitting details biophysical neuron models to electrode recording is very hard open problem. Solving this problem is timely considering that more organoids with precise read-write electrode arrays are becoming available. It is also a search space in between single unit data fitting, and large scale spike recordings where there is no consensual method. Research in this space is important and innovative.

2) The model proposed makes a lot of sense in terms of biophysics, the description is clear and meaningful for the nature of the data.

3) The optimization model combines gradient descent in a biophysical non-linear neuron model which is notoriously very hard and unstable. It is a remarkable achievement that it is even possible.

Weaknesses:
1) I believe that a reminder about the definition of SBI would have been very useful for the reader in the main text. I saw the recall on SBI from the appendix but I could not find the specification of the q_phi network. What does it take as inputs (the feature spaces? or the voltage traces?) and how does it model the posterior output (deep mutimodal gaussian distribution?).

2) The model is simulated with Jax therefore offering gradient based optimization in the forward biophysical model. At first I thought, that the paper used an algorithm which combines differentiable forward model with SBI. But now I believe that it is one or the other. But it would be great to clarify this upfront to avoid any possible confusion.

3) The authors conclude somewhat rapidly that gradient descent is less good than SBI (Figure 3 C,D,E and line 290-305).  Unfortunately the data fitting of non-linear model via gradient descent not fully validated, and the validation process is not well documented. Gradient descent in jax for HH model is a great innovation, but without more reports on the intermediates validation steps, I find it very plausible that the gradient optimization was not well implemented. It is certainly a subject on it's own but here are a few possibilities why gradient based optimization might have been implemented without care (if those things were not addressed, it might be wise to mention some of them as limitations of the implemented gradient descent approach):

3a) HH is a complex non-linear system where the gradients are likely to be vanishing and exploding at many steps of the differential equation simulation.

3b) The stability of the forward simulation, and of the gradients, are likely to depend on the ODE integration that is being used.

3c) The gradient optimization with Adam is not invariant to a non-linear reparametrization of the biophysical parameters (like conductances), so there might exist some non-linear reparametrization of the system towards a more friendly and stable model.

3d) Because of the problems above, spiking models optimized to fit single units [1] and large networks [2] via gradient descent are usually  parameterized using linear filters (GLM, conv1d) or as current based LIF models rather than non-linear conductance based model. I suspect that it would possible to implement a multi-compartment model of this type and build an even more stable gradient descent method to recover the stimulation sources for instance (even if the model is not biophysically interpretable like HH).

3e) It would be possible to integrate the parameter priors into the gradient descent optimization. Basically, via simple langevin dynamics where after each gradient descent step on the likelihood,  one does a gradient descent step on the parameter prior. It is not clear whether it was used here.

4) In Figure 4, rather than a scatter plot it would be great to see a density with estimated parameters. Because if we want to judge to plausibility of the output distribution we need to see whether the peak of the distribution is landing is a biologically plausible range.

5) I would have wished to see more information about the real dataset and how the task is defined. How many minutes of recording is available? I understand that there is no notion of training set and test set which is somewhat unconventional for gradient descent problem. I see how the problem is probably well posed anyway though. Yet a more detailed description of the inference problem could have been useful.

6) It is somewhat unusual to put the related version before discussion in this conference. The writing exercise is somewhat different when these references come after the introduction, and the paper becomes often less shiny. However, I do not want to impose this writing style so I do not take this into account in my grade.

7) In Table 1, it would be great to show with arrows which measure is improved with higher or lower values.

[1] GLM models (see https://www.nature.com/articles/nature07140)
[2] Recurrent networks of LIF neurons https://elifesciences.org/articles/106827

---

> ### Author Rebuttal · Authors · 2026-03-31
>
> We thank the reviewer for the thorough and insightful review! We address each point below.
>
> **[W1 — SBI network specification]** The density estimator $q_\phi(\theta \mid x)$ takes as input the same engineered feature vector used for gradient-based fitting (peak amplitudes, timing features, stimulus thresholds), not raw voltage traces. The posterior over biophysical parameters $\theta$ is modeled as a gaussian distribution using a normalizing flow, implemented via amortized NPE in the sbi library [Tejero-Cantero et al., 2020]. We will add a concise summary to the main text and SBI appendix section in the final version if accepted.
>
> **[W2 — GD and SBI are separate pathways]** The two methods are indeed independent alternatives: gradient descent explicitly exploits JAXLEY's differentiability to optimize a deterministic forward model, while SBI uses it purely as a forward simulator without backpropagating through it. Importantly, SBI is designed for inference in stochastic models with random noise currents — in that setting, the marginal likelihood is intractable and gradient-based optimization is not directly applicable. We will clarify this distinction upfront in the main text.
>
> **[W3 — Gradient descent validation]** To clarify, we do not intend to conclude that gradient descent is generally worse than SBI. We will rewrite lines 290–305 to prevent this confusion, and will add more detail to the gradient descent implementation section Appendix E. Figure 3D/E shows that on noiseless simulated data, GD and SBI perform equivalently. GD only underperforms SBI in Figure 3D/E when noise is injected into the forward simulator, because GD optimizes a misspecified model with no mechanism to account for the latent injected-current noise variable, whereas SBI explicitly models it. Regarding points (3a)–(3e): (3a/3b) the JAXLEY framework [Deistler et al., 2025] we build upon includes a systematic study of optimization stability for differentiable HH simulation. (3c) All parameters are reparameterized via sigmoid transforms into physiological bounds, which substantially stabilizes optimization. (3d) We agree that multicompartment version of GLM or LIF could in principle be used to predict multi-electrode stimulation responses and would be an interesting direction for future work. A key advantage of our biophysical HH framework, however, is that it can in principle predict responses to arbitrary stimulus waveforms, durations, and shapes, not just the triphasic pulses used in our experiments, since the forward model is grounded in membrane dynamics rather than fitted to a specific stimulus class. Whether a GLM or LIF surrogate based model could retain this generality is an open question we leave for future work. (3e) We do not use Langevin dynamics explicitly, but prior knowledge is incorporated through: (i) hard physiological bounds enforced by the sigmoid reparameterization, equivalent to a uniform prior, and (ii) explicit smoothness penalties on axon trajectory geometry in the loss.
>
> **[W4 — Figure 4: density plots]** Figure 4 shows predicted vs. observed *features* rather than the inferred parameters themselves. We note that Appendix P already includes posterior marginal distributions of inferred parameters for SBI (Figures 18 and 19), which shows that inferred parameter values cluster in biologically plausible ranges. For the gradient descent results, we will add a figure showing marginal distributions of inferred parameters and will include this in the main text space permitting.
>
> **[W5 — Dataset details]** This is an excellent point and we will add a dedicated dataset appendix in the final version. Briefly: EIs were extracted from ~30 min of light-evoked activity (though less suffices in principle). Single-electrode threshold measurement across all cells and electrodes in a preparation required ~90 min (see response to Reviewer DdHr). Multi-electrode stimulation responses for a single cell across one electrode triplet required ~3 hours.
>
> **[W6 — Related work placement]** We note this is a deliberate structural choice but will consider restructuring if space and flow allow.
>
> **[W7 — Table 1 directional arrows]** We will reformat Table 1.
>
> **[Key Question — GD vs SBI discrepancy between Fig. 3 and Table 1]** The apparent contradiction is partially explained by the distinction between noise and model misspecification. In Figure 3, the stochastic forward model is well-specified and SBI correctly accounts for injected current noise, giving it a structural advantage. In Table 1, real tissue introduces model misspecification: SBI relies on simulated training data faithfully representing real observations and is therefore more sensitive to distributional shift. GD is also affected by misspecification (evidenced by degraded feature recovery in Appendix N) but it is more robust for the downstream multielectrode stimulus prediction task. This is an important result we will highlight more clearly.

---

> > ### Author Rebuttal · Reviewer_h6Bp · 2026-04-02
> >
> > Thank you. The responses from the author are clear. I believe that my initial judgement was fair.

---

### Review · Ethics_Reviewer_speM · 2026-03-30

**Recommendation:** Remediation action needed

**Ethics Issue:**

This is a strong paper. From an ethical perspective, there are no hugely problematic issues. However, the paper would overall benefit from clearer boundary-setting. For example, the experiments are conducted on ex vivo macaque retina, yet the manuscript emphasizes precise neurostimulation and time-saving clinical relevance. I would encourage the authors to state more explicitly what has and has not been validated, especially with respect to future human applications. A sentence on the matter should be enough.

A second issue is potential over-trust in the inferred model. Because the paper presents a biophysically grounded “digital twin” that predicts unseen stimulation responses, readers may infer a level of fidelity that exceeds what has actually been shown. The authors should therefore clarify where the model may fail, what uncertainty remains, and why success in this ex vivo setting does not yet justify broader reliance for stimulation planning.

Third, the impact discussion is currently too minimal relative to the application domain. A brief but more specific statement addressing translational limits, uncertainty, and future closed-loop use would strengthen the paper. If not already included elsewhere, I would also welcome a short statement on macaque tissue sourcing and ethical approval.

**Remediation Action:**

See above. The main points are:
- clearer boundary setting
- clarification of where the models might fail and lack of justification for broader reliance
- revised impact statement
- (if available) information on ethical approval and macaque tissue sourcing

---

### Decision · Program_Chairs · 2026-04-30

**Decision:**

Accept (spotlight)

**Comment:**

This work presents a simulation-based approach for predicting the effects of targeted neurostimulation. Specifically the authors present a way to use multi-electrode array recordings to infer the biophysical model of the network, and then use in silico simulation to test different stimulation patterns. The work is very timely-- targeted neurostimulation is a fast-growing area of interest and this work presents a set of novel computational tools to approach this challenge with. There were some more minor concerns of the reviewers, however these were primarily clarification questions that were answered satisfactorily by the authors. Given the importance of the work and the positive assessment of the reviewers, I recommend this work to be accepted.